# Structure of the Nmd4-Upf1 complex supports conservation of the nonsense-mediated mRNA decay pathway between yeast and humans

**Irène Barbarin-Bocahu[1], Nathalie Ulryck[1], Amandine Rigobert[1], Nadia Ruiz Gutierrez[2], Laurence Decourty[3], Mouna Raji[1], Bhumika Garkhal[1], Hervé Le Hir[2], Cosmin Saveanu[3], Marc Graille ⓘ[1] ***

**1** Laboratoire de Biologie Structurale de la Cellule (BIOC), CNRS, Ecole polytechnique, Institut Polytechnique de Paris, Palaiseau, France, **2** Institut de Biologie de l'Ecole Normale Supérieure (IBENS), Ecole Normale Supérieure, CNRS, INSERM, PSL Research University, Paris, France, **3** Institut Pasteur, Université Paris Cité, Unité Biologie des ARN des Pathogènes Fongiques, Paris, France

* marc.graille@polytechnique.edu

**Data Availability Statement:** All relevant data are within the paper and its Supporting information files, except for coordinates and structure factors

## Abstract

The nonsense-mediated mRNA decay (NMD) pathway clears eukaryotic cells of mRNAs containing premature termination codons (PTCs) or normal stop codons located in specific contexts. It therefore plays an important role in gene expression regulation. The precise molecular mechanism of the NMD pathway has long been considered to differ substantially from yeast to metazoa, despite the involvement of universally conserved factors such as the central ATP-dependent RNA-helicase Upf1. Here, we describe the crystal structure of the yeast Upf1 bound to its recently identified but yet uncharacterized partner Nmd4, show that Nmd4 stimulates Upf1 ATPase activity and that this interaction contributes to the elimination of NMD substrates. We also demonstrate that a region of Nmd4 critical for the interaction with Upf1 in yeast is conserved in the metazoan SMG6 protein, another major NMD factor. We show that this conserved region is involved in the interaction of SMG6 with UPF1 and that mutations in this region affect the levels of endogenous human NMD substrates. Our results support the universal conservation of the NMD mechanism in eukaryotes.

## Introduction

In eukaryotes, several cytoplasmic surveillance pathways monitor the quality of translated mRNAs to prevent ribosomes stalling on faulty mRNAs or the synthesis of aberrant proteins [1–4]. Among these, the nonsense-mediated mRNA decay (NMD) pathway is responsible for the rapid detection and degradation of translated mRNAs harboring premature termination codons (PTCs), which can arise from mutations, transcription, or processing errors as well as alternative or defective splicing events [3–6]. The NMD mechanism also targets mRNAs, small nucleolar RNAs (snoRNAs), and long noncoding RNAs (lncRNAs) carrying normal stop codons located in a specific context (short upstream open reading frame or uORF, long 3′

for the two structures described. The related files are already available in the PDB website using entry codes 8RDD and 8RD3.

**Funding:** This work was supported by the Centre National de la Recherche Scientifique (CNRS) to HLH and MG, the Ecole Normale Supérieure and the Institut National de la Santé et de la Recherche Médicale for HLH, Institut Pasteur to CS, the Agence Nationale pour la Recherche ANR-18-CE11-0003-01 to CS, ANR-18-CE11-0003-02 and ANR-22-CE12-0004 to HLH and ANR-18-CE11-0003-04 to MG, Ecole Polytechnique to MG, the French Ministère de l'Enseignement Supérieur et de la Recherche (MESR) to IBB/BG, Ecole doctorale Complexité du Vivant (ED 515, Sorbonne Université) for PhD funding of NRG and the Fondation ARC pour la Recherche sur le Cancer for PhD funding of NRG and IBB. The funders had no role in study design, data collection and analysis, decision to publish, or preparation of the manuscript.

**Competing interests:** The authors have declared that no competing interests exist.

**Abbreviations:** EJC, exon junction complex; HD, helicase domain; ITC, isothermal titration calorimetry; lncRNA, long noncoding RNA; NMD, nonsense-mediated mRNA decay; PTC, premature termination codon; RQ, relative quantification; snoRNA, small nucleolar RNA; uORF, upstream open reading frame.

UTRs, low translational efficiency or exon-exon junction located downstream of a stop codon [3,4,7–13]). NMD hence plays an important role in gene expression regulation and could monitor that the correct reading frame is scanned by the ribosome since most out-of-frame translation events would result in the detection of a premature stop codon by the ribosome [14]. The NMD pathway is commonly considered as a double-edged sword protecting cells from the synthesis of potentially harmful truncated proteins but also inhibiting the synthesis of partially or fully functional truncated proteins. Thereby, NMD is considered to be involved in many forms of cancers [15,16], in about 20% of all genetic diseases caused by nonsense mutations such as neurodegenerative diseases [17,18] and also to be a major defense mechanism against viral infections [19,20].

NMD occurs through successive steps aimed at recognizing a stop codon as premature and then at subsequently degrading the aberrant mRNAs. The precise molecular mechanism of the NMD pathway is still unclear and remains controversial. Many lines of evidence supported the existence of several possible scenarios to recruit different factors to trigger the decay of NMD substrates [21]. Two models, which are not mutually exclusive, are currently proposed for NMD. Both rely on the involvement of 3 particularly important factors, UPF1, UPF2, and UPF3, in various eukaryotic model systems. The first model, the 3′-*faux* UTR model posits that for mRNAs with long 3′ UTRs, a long spatial distance between a stop codon and the mRNA poly(A) tail destabilizes NMD substrates. Indeed, it would prevent the physical interaction between the eRF1-eRF3 translation termination complex recognizing a stop codon in the A-site of the ribosome and the poly(A)-binding protein (Pab1 or PABP in *Saccharomyces cerevisiae* and human, respectively) bound to the 3′ poly(A) tail [22–24]. In this context, the UPF1 RNA helicase interacts with the eRF1-eRF3 complex bound to terminating ribosomes together with the UPF2 and UPF3 proteins [25,26], thereby tagging this stop codon as a PTC, and subsequently recruits RNA decay factors [27]. Alternatively, for mRNAs harboring PTCs more than 50 to 55 nucleotides upstream of an exon-exon junction [28], the presence of the exon junction complex (EJC) bound downstream of the PTC triggers mRNA elimination by the NMD pathway. Indeed, the recognition of PTCs by the SURF complex (for SMG-1/UPF1/eRF1-3 complex) allows the interaction of this complex with the UPF2 and UPF3 proteins bound to the EJC, thereby leading to the formation of the DECID complex (for Decay Inducing Complex) and the subsequent phosphorylation of UPF1 [26]. The phosphorylated UPF1 can then recruit the SMG5/SMG7 heterodimer, which subsequently interacts with the CCR4-NOT deadenylation complex and the decapping factors to eliminate aberrant mRNAs [29,30]. Alternatively, the SMG6 endonuclease can be recruited via its interaction with the EJC or through phospho-dependent and phospho-independent interactions with UPF1 [31–37].

Recent publications have provided some nuances to these initial models, calling for further studies aimed at clarifying the molecular mechanism of NMD. First, *S. cerevisiae* Pab1 is dispensable to discriminate between normal and premature stop codons [38]. Second, the EJC now emerges as an enhancer of NMD even if its interaction with UPF3 is not mandatory for this process [24,39,40]. This can be explained for example, by the role of EJC in enhancing translation [41], which, in turn can facilitate the detection and degradation of NMD substrates. Third, human UPF3B interacts directly with UPF1 as well as with eRF1 and eRF3a, unlike UPF1 [42], in opposition with the SURF model. Fourth, the SMG6-dependent RNA degradation pathway in NMD depends on the SMG5/7 complex, as the absence of the latter prevents endonucleolytic cleavage of mRNA by SMG6 [43]. This is in line with a previous observation showing that SMG6 and SMG5-7 complexes target essentially the same genes [44]. Finally, using fast affinity purification coupled to quantitative mass spectrometry [45], we recently showed that in the yeast *S. cerevisiae*, Upf1 is part of 2 distinct and mutually exclusive complexes: the Upf1-2/3 complex and the Upf1-decapping complex. The latter encompasses Upf1,

the Dcp1, Dcp2, and Edc3 decapping factors as well as 2 largely uncharacterized proteins, Nmd4 and Ebs1. These 2 additional yeast NMD co-factors show similarities with metazoan SMG6 and SMG5/7, both at the level of their sequence and in term of their interactions with Upf1.

UPF1 is made of 2 structured domains: a CH-domain known to interact with UPF2 and an ATP-dependent RNA helicase domain (HD). Both domains are surrounded by N- and C-terminal extensions, which are predicted to be unfolded. Ebs1 is predicted to contain a 14-3-3 domain [46]. Such domain is generally involved in the recognition of phosphoserine or phosphothreonine residues [47]. Ebs1 acts as a translation inhibitor and promotes NMD [46,48] but no other studies have focused on this protein. Nmd4 is composed of a PIN domain (for PilT N-terminus) [49] and interacts directly with the helicase domain of Upf1 [45]. Although Nmd4 is not an essential NMD factor, it plays an important role for NMD under specific conditions such as the presence of a Upf1 protein lacking its CH domain [45]. Interestingly, yeast Ebs1 could be a functional homolog of metazoan SMG5, SMG6, and SMG7 proteins, all of which include a 14-3-3 domain [31,46,50,51], while the PIN domain of Nmd4 is structurally similar to the PIN domains of SMG5 and SMG6 [34,49]. Overall, these studies suggest that a global mechanism is conserved from yeast to human for the NMD pathway, with some elaborate branches (such as the role of the EJC) in metazoans. The truly conserved features of such a universally conserved molecular mechanism of NMD are unclear.

Here, we describe the crystal structure of the complex formed between Nmd4 and the Upf1 helicase domain of *S. cerevisiae*. This structure reveals that Nmd4 interacts with Upf1 helicase core through a conserved C-terminal arm, an interaction that is important for the NMD-activating role of Nmd4 in yeast. We also reveal that Nmd4 activates the ATPase activity of Upf1 and increases its affinity for RNA. Interestingly, we identify a region similar to the C-terminal arm of Nmd4 in SMG6 proteins in metazoans. We show that this region is important for the previously described phospho-independent interaction of the human SMG6 protein with the helicase domain of UPF1, as well as for the optimal degradation of NMD substrates in human cells. These results suggest that, in addition to the endonuclease function described for SMG6, which appears to have been lost in Nmd4, the interaction of this protein with UPF1 may itself be important for NMD through its impact on the RNA helicase properties of this central NMD factor.

## Results

### Nmd4 wraps around Upf1 helicase domain mainly through its C-terminal domain

Given that the yeast Upf1 helicase domain (Upf1-HD, amino acids 221 to 851) interacts directly with Nmd4 [45], we purified the 2 proteins individually and reconstituted the Nmd4/Upf1-HD complex. We tried to crystallize this complex in the absence or presence of a short RNA oligonucleotide, but only obtained crystals of the complex in the absence of RNA. The crystal structure of this complex was solved to 2.4 Å resolution (Fig 1A and 1B and S1 Table). We also determined the 1.8 Å resolution crystal structure of the PIN domain of Nmd4 (residues 1 to 167) to help us solve the structure of the Nmd4/Upf1-HD complex. As this structure is virtually identical to the structure of the PIN domain of Nmd4 in the complex (rmsd of 0.5 Å over 163 Cα atoms between the 2 structures), we will only describe the structure of this domain in the Upf1-Nmd4 complex. In the Nmd4/Upf1-HD complex, Upf1-HD adopts a conformation similar to that of human UPF1-HD bound to UPF2 or ADP and phosphate, with the largest difference observed in the orientation of domain 1B (rotation of this domain by 17 to 19° relative to the RecA1 domain in both cases; S1A and S1B Fig; [52,53]). The Nmd4

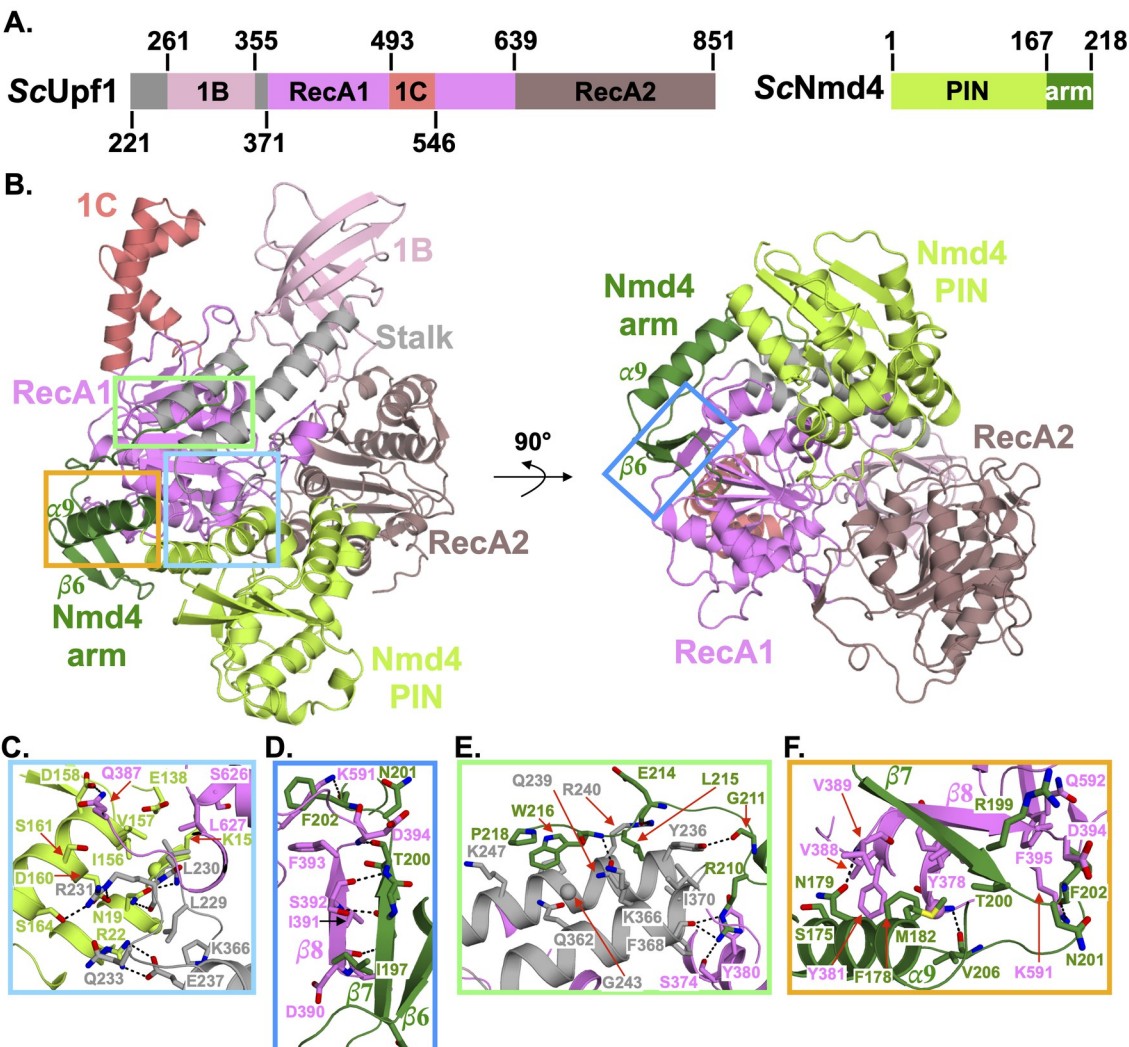

**Fig 1. The Nmd4 protein wraps around Upf1 RecA1 domain.** (A) Schematic representations of *S. cerevisiae* Upf1 helicase domain and Nmd4 protein with domain boundaries indicated. The different protein domains are shown with a color code, which is used in all the figures. (B) Cartoon representation of the structure of the Nmd4/Upf1-HD complex. The left and right orientations are related by a 90° rotation along the horizontal axis. (C–F) Zoom-in on 4 regions (boxed in panel B) of the Nmd4/Upf1-HD interface. Hydrogen bonds and salt bridges are depicted as black dashed lines. HD, helicase domain.

protein is composed of 2 domains. First, a PIN domain (residues 1 to 167; Fig 1B), which is structurally similar to the PIN domains of *Kluyveromyces lactis* Nmd4 (*Kl*Nmd4; S1C Fig; [49]) and of the human proteins SMG5 and SMG6 (S1D and S1E Fig; [34]). As previously observed for *Kl*Nmd4 [49], the catalytic residue D1353 of the human SMG6 protein corresponds to a leucine (Leu112) in the *S. cerevisiae* Nmd4 protein (S1F Fig). Since the D1353A mutation completely abolishes the enzymatic activity of SMG6 [34], this strongly suggests that the PIN domain of Nmd4 is not endowed with endonucleolytic activity. This is in line with the lack of evidence for such activity in the yeast NMD pathway, in contrast to what has been shown in metazoans [32,33]. Second, a C-terminal region (hereafter referred to as the « arm »; residues 168 to 218) folds into a long α-helix (α9) followed by a two-stranded antiparallel β-sheet (strands β6 and β7) and a long unstructured region (residues 202 to 218) to embrace

Upf1-HD. Interestingly, in the structure of *Kl*Nmd4 determined in the absence of Upf1-HD, only a small fraction of this « arm » could be modeled; however, it adopts a radically different structure, suggesting intrinsic flexibility in the absence of Upf1 (S1C Fig).

The overall quality of the electron density map allowed us to unambiguously identify the residues of the 2 proteins involved in the formation of the complex (S2A and S2B Fig). In total, 39 residues of Nmd4 interact with 44 residues located in the RecA1 domain and in the stalk of Upf1-HD to form an interface area of 2,050 Å$^2$ (Figs 1B, S3 and S4; [54]). These residues are relatively well conserved in the eukaryotic proteins Nmd4 and Upf1 (S3 and S4 Figs). At the interface, 2 salt bridges are formed between R22 and D160 from Nmd4 PIN domain and E237 and R231 from Upf1-HD stalk region, respectively (Fig 1C). Twenty-one hydrogen bonds are also involved in complex formation (listed in S2 Table). Among these, 4 are formed between amino acids of the PIN domain and R231 of the Upf1-HD stalk region (Fig 1C). Another hydrogen bond is formed between D158 of the PIN domain and Q387 of the RecA1 domain of Upf1 (Fig 1C). All other hydrogen bonds originate from residues on the Nmd4 « arm » (S2 Table). In particular, the main chain atoms from N198 and T200 of the strand β7 of Nmd4 form hydrogen bonds with D390, S392, and D394 main chain atoms from Upf1 RecA1 domain (Fig 1D). As a result, a parallel β-zipper is formed between strand β7 of Nmd4 and strand β8 of Upf1 at the interface, generating a three-stranded β-sheet common to the RecA1 domain of Upf1 and the « arm » of Nmd4. Another Nmd4 residue important for the interface is the strictly conserved R210 (S3 Fig), as its side chain forms 3 hydrogen bonds with Upf1 residues (Fig 1E). Finally, several residues in the C-terminal « arm » of Nmd4 form hydrogen bonds via their main chain atoms with Upf1 residues (Fig 1E and 1F and S2 Table). In addition to these electrostatic interactions, the side chains of several hydrophobic residues contribute to the interface. In Nmd4, these residues mainly belong to the « arm ». Indeed, F178 and M182 of Nmd4 interact with Y378, Y381, V388, and I391 of Upf1 RecA1 domain (Fig 1F). Similarly, the strictly conserved residues L215, W216, and P218 of the Nmd4 « arm » are positioned at the interface between the 2 helices of the Upf1 stalk region (Figs 1E and S3). In conclusion, our crystal structure reveals that Nmd4 interacts primarily with 2 structurally important domains of Upf1-HD, namely the RecA1 and stalk domains, and to a lesser extent with the RecA2 domain.

## Nmd4 stimulates Upf1 ATPase activity

In the Nmd4/Upf1-HD interface, the Nmd4 « arm » accounts for around 75% of the interface area, while its PIN domain represents only 25%. This led us to investigate the importance of each Nmd4 domain for the interaction with Upf1-HD. Using in vitro His-pull down, we observed that Upf1-HD interacts specifically with full-length Nmd4 or with the « arm » region alone, but not with the PIN domain (S5A Fig). This interaction was further quantified by iso-thermal titration calorimetry (ITC) experiments showing that full-length Nmd4 or Nmd4 « arm » interacted with Upf1-HD with a Kd of 2.1 μM and 1.96 μM, respectively, whereas no interaction was detected between the PIN domain of Nmd4 and Upf1-HD in our experimental conditions (Table 1 and Fig 2A). In parallel, we performed co-immunoprecipitation experiments in vivo. Unfortunately, an HA-tag version of the « arm » was not stably expressed in *S. cerevisiae* yeast. However, unlike the full-length Nmd4 protein, the single PIN domain did not co-purify with TAP-tagged Upf1 (S5B Fig). Overall, these results indicate that the « arm » region of Nmd4 is sufficient and necessary for a stable Nmd4/Upf1 interaction.

As the role of the interaction between Nmd4 and Upf1 is still largely unknown, we analyzed its effect on Upf1 ATPase activity. We observed that, in vitro, the basal ATPase activity of the yeast Upf1-HD was strongly stimulated upon incubation with the full-length Nmd4 protein

**Table 1. Thermodynamics parameters for the interaction of Upf1-HD with Nmd4 or RNA as determined by ITC.**

| Receptor | Ligand | Stoichiometry ($n$) | Kd ($\mu$M) | $\Delta$H (kcal/mol) | T$\Delta$S (kcal/mol) |
|---|---|---|---|---|---|
| Upf1-HD | His$_6$-ZZ-Nmd4 FL | 1.08 ± 0.15 | 2.1 ± 0.56 | −14.85 ± 0.34 | −7.21 ± 0.37 |
| Upf1-HD | His$_6$-ZZ-Nmd4 « arm » | 0.98 ± 0.06 | 1.96 ± 0.11 | −12.45 ± 2.2 | −4.78 ± 2.18 |
| Upf1-HD | His$_6$-ZZ-Nmd4 PIN | nd | nd | nd | nd |
| Upf1-HD | His$_6$-ZZ | nd | nd | nd | nd |
| Upf1-HD | RNA poly(U)$_{30}$ | 0.59 ± 0.08[a] | 1.03 ± 0.28 | −34.97 ± 7.02 | −26.86 ± 7.14 |
| His$_6$-ZZ-Nmd4 FL/Upf1-HD | RNA poly(U)$_{30}$ | 0.47 ± 0.02[a] | 0.44 ± 0.21 | −55.79 ± 2.28 | −47.17 ± 2.29 |
| His$_6$-ZZ-Nmd4 FL | RNA poly(U)$_{30}$ | 0.66 ± 0.12[a] | 8.1 ± 2.95 | −14.02 ± 0.91 | −7.16 ± 1.14 |
| His$_6$-ZZ | RNA poly(U)$_{30}$ | nd | nd | nd | nd |

nd: not detectable ($n$ = 2).

Mean and standard deviation values calculated from 3 replicates.

[a] For the titrations of RNA poly(U)$_{30}$ with either Upf1-HD, Nmd4 or the complex, we observed stoichiometries lower than 1, indicating that up to 2 proteins can bind to a single poly(U)$_{30}$ RNA. For Upf1-HD, this is in agreement with previous structural data, which revealed that Upf1 binds a stretch of 9 Us [55].

HD, helicase domain; ITC, isothermal titration calorimetry.

(Fig 2B). The « arm » of Nmd4, which is mandatory for the Nmd4/Upf1-HD interaction, also enhanced Upf1-HD ATPase activity, but to a much lower extent. Surprisingly, the PIN domain alone also stimulated Upf1-HD helicase activity as effectively as the « arm ». The various Nmd4 constructs alone had no ATPase activity. Similarly, the His-ZZ tag used to express the various Nmd4 construct did not activate significantly Upf1-HD activity. This indicates that the observed increase in enzymatic activity is specific to the binding of Nmd4 to Upf1-HD (Fig 2B). Altogether, this strongly suggests that the PIN domain and the « arm » act synergistically to enhance Upf1 ATPase activity, but also that a physical interaction between Upf1-HD and the PIN domain might exist in vitro. We were unable to detect such an interaction using our different interaction assays (pull-down and ITC), which are optimal for studying interactions with dissociation constants (Kd) in the nanoM to tens of microM range. We therefore assume that a transient low-affinity interaction (high Kd value not detected by our binding assays) exists between Upf1-HD and PIN Nmd4 and can only be detected by highly sensitive assays such as our radioactivity-based ATPase assay, which was performed with a 20-fold molar excess of PIN Nmd4 domain over Upf1-HD. This also indicates that Nmd4 has a long-range effect on Upf1 helicase activity, since no Nmd4 residue interacts in the vicinity of the Upf1 ATPase active site. This could be due to the extended interaction of Nmd4 with 3 important regions (RecA1, stalk, and RecA2) of Upf1-HD.

To investigate the influence of Nmd4 on another Upf1-HD activity, namely RNA binding, we determined the Kd of Upf1-HD for a poly(U)$_{30}$ oligonucleotide in the absence or in the presence of Nmd4 by ITC. First, Upf1-HD, but also surprisingly Nmd4 alone, interacted with a poly(U)$_{30}$ RNA with Kd values of 1.03 $\mu$M and 8.1 $\mu$M, respectively (Table 1 and S6 Fig). Interestingly, the Nmd4/Upf1-HD complex has a 2.3-fold lower Kd value for this RNA than Upf1-HD (Kd = 0.44 $\mu$M). Whether this increase in affinity is due to a synergistic effect between both proteins or to an allosteric stimulation of one partner on the RNA binding property of the second partner remains to be clarified.

## The Nmd4/Upf1-HD interaction is important for NMD in vivo

Detailed analysis of the crystal structure of the Nmd4/Upf1-HD complex revealed that 2 strictly conserved residues, located in the « arm » region of Nmd4, were of particular interest. The side chain of R210 forms 3 hydrogen bonds with the carbonyl group of F368 as well as

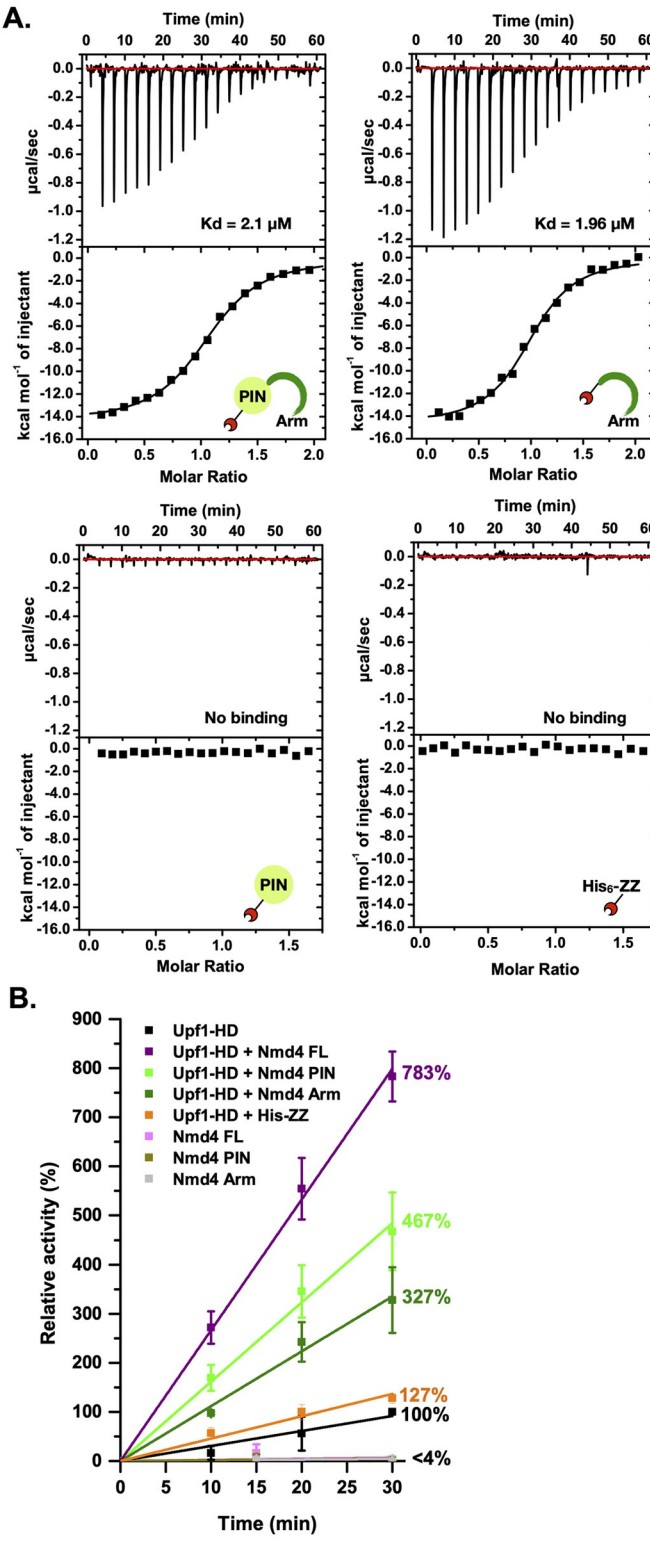

**Fig 2. Nmd4 influences Upf1 ATPase activity.** (A) Characterization of the interaction between His$_6$-ZZ-Nmd4 full-length protein (upper left panel), His$_6$-ZZ-Nmd4 « arm » (upper right panel), His$_6$-ZZ-Nmd4 PIN domain (lower left panel) or the His$_6$-ZZ tag (lower right panel), and Upf1-HD by ITC. Upper panel: ITC data obtained by injecting the different Nmd4 constructs or the His$_6$-ZZ tag into Upf1-HD. Lower panel: Fitting of the binding curves using a single binding site model. Three biologically independent experiments were performed and a representative image is shown.

The data underlying this figure can be found in S1 Data. (B) Nmd4 stimulates Upf1-HD ATPase activity. Purified Upf1-HD alone or in combination with full-length or fragments of Nmd4 or His-ZZ (control) was used in a $^{32}$P release assay, as a readout of the ATPase activity. The relative activity (indicated in percent) corresponds to the activity measured in a given condition (different proteins) normalized to the activity of Upf1-HD alone after 30 min of reaction at 30°C. The relative activity for each condition after 30 min is indicated on the right of the different curves. The data underlying this figure can be found in S2 Data. HD, helicase domain; ITC, isothermal titration calorimetry.

with the hydroxyl group of S374 from Upf1-HD, while the W216 side chain stacks on the helix α1 and in particular on G243 of the Upf1-HD stalk region (Fig 3A and 3B and S2 Table). To investigate the importance of these residues for complex formation, we generated 2 single point mutants (R210E to introduce charge inversion and W216A to disrupt several van der Waals interactions) as well as the double mutant (R210E/W216A; referred to as M2). Similarly, we mutated Upf1-HD G243 and G377 to arginines to induce steric hindrance with Nmd4 as they lie close to residues W216 and R210 of Nmd4, respectively (Fig 3A and 3B). Again, 2 single point mutants (G243R or G377R) as well as the double mutant (G243R/G377R, referred to as DM) were generated. During the purification process, all mutants behaved similarly to the wild-type proteins, indicating that the overall structure of the proteins were not affected by the mutations (S7A and S7B Fig). All Upf1-HD mutants lost their ability to interact in vitro with Nmd4 compared with the wild-type protein as demonstrated by the specific enrichment of Upf1-HD WT but not of the mutants by CBP-His$_6$-ZZ-Nmd4 (compare lane 7 to lanes 8 to 10; Fig 3C). Thus, the substitution of glycine by a large, positively charged arginine side chain strongly interferes with the formation of the Nmd4/Upf1-HD complex. Similarly, we observed that each Upf1 single point mutant (G243R or G377R) associated much more weakly with Nmd4-HA in vivo (Fig 3D). We then studied the effect of these mutants on the in vivo stability of 2 endogenous (DAL7 and pre-L28) yeast NMD substrates in a specific context where Nmd4 is known to be important for NMD [45]. The expression of a Upf1 protein lacking the CH domain (hereafter referred to as Upf1-HD-Ct) only partially complemented the UPF1 deletion compared with ectopic expression of the full-length protein (pUpf1; Figs 3E and S8; [45]). Importantly, the complementation by Upf1-HD-Ct is further reduced in the *upf1Δ/nmd4Δ* double mutant. In the presence of the Upf1-HD-Ct construct, the levels of DAL7 and pre-L28 NMD substrates were similar in the *upf1Δ* strains lacking Nmd4 or upon expression of Upf1-HD-Ct G243R or G377R point mutants in the *upf1Δ* or *upf1Δ/nmd4Δ* strains. Altogether, this indicates that the interaction between Nmd4 and Upf1 is important for NMD in the absence of the Upf1 CH domain, i.e., when the efficient recruitment of Upf2 and/or decapping factors is impaired for instance [27,53].

We also observed that the various Nmd4 mutants (R210E, W216A, and M2) were strongly affected in their ability to interact with Upf1-HD compared with the wild-type Nmd4 protein in vitro (Fig 3F), confirming the role of the Nmd4 « arm » in the interaction with Upf1-HD. Consistent with these observations, these 3 Nmd4 mutants did not stimulate Upf1 ATPase activity as efficiently as the wild-type Nmd4 protein (Fig 3G).

Overall, these experiments show that the interaction formed between the Nmd4 « arm » and Upf1-HD contributes to Upf1 function in NMD, and in particular to its ATPase activity.

## An arm-like region of human SMG6 contributes to its binding to UPF1 and its role in NMD

The full-length SMG6 protein interacts with the helicase domain of human UPF1 in a phospho-independent manner [31,36,37]. This interaction depends on the presence of the UPF1-HD stalk region [36], which also plays a central role in the interaction between *S*.

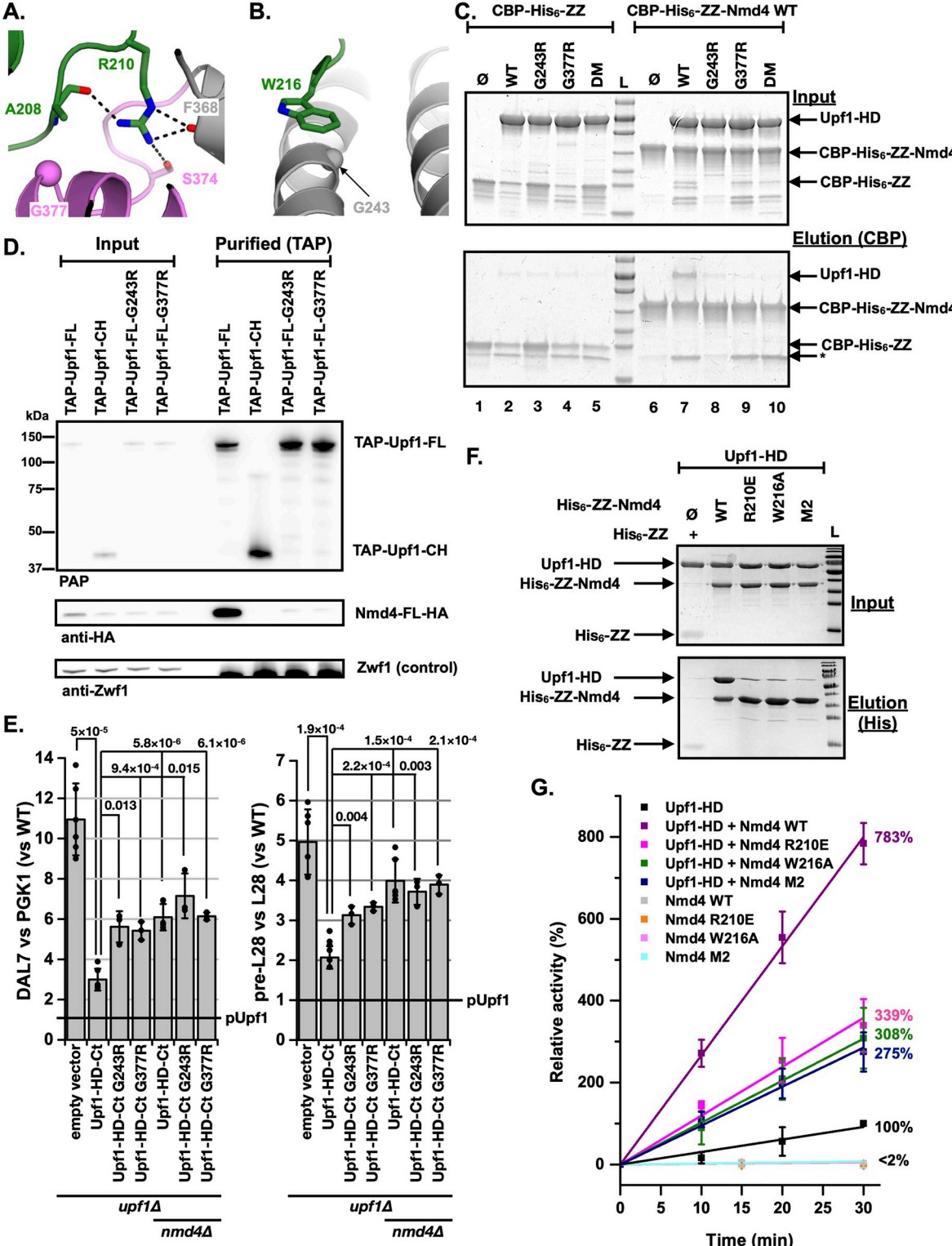

**Fig 3. The interaction of Nmd4 with Upf1 is important for NMD.** (A) Zoom-in on the interaction network formed by the side chain of Arg210 from Nmd4. Hydrogen bonds are depicted as black dashed lines. The Cα atom of Gly377 from Upf1-HD is shown as a sphere. (B) Zoom-in on the side chain of Trp216 from Nmd4 lying on helix α1 from Upf1-HD stalk region. The Cα atom of Gly243 from Upf1-HD is shown as a sphere. (C) CBP-pull-down experiment showing that the Upf1-HD single mutants (G243R or G377R) or the double mutant (DM; G243R/G377R) affect the binding to CBP-tagged Nmd4. Input and eluate from calmodulin sepharose beads (IP) samples were

analyzed on 12% SDS/PAGE and Coomassie blue staining. As controls, each Upf1-HD variant was incubated with the CBP-His$_6$-ZZ tag, revealing some weak nonspecific retention of the various Upf1-HD proteins on the calmodulin sepharose beads. Degradation products enriched on calmodulin sepharose beads are indicated by an asterisk (*). Three biologically independent experiments were performed and a representative image is shown. (D) Single point mutations in full-length Upf1 disrupt its interaction with Nmd4 in vivo. Co-purification of Nmd4-HA with TAP-Upf1-FL, TAP-Upf1-FL-G243R, and TAP-Upf1-FL-G377R was evaluated by immunoblot. A negative control experiment used the N-terminal region of Upf1 (TAP-Upf1-CH, encompassing residues 1 to 208) known to be unable to interact with Nmd4 [45]. (E) Upf1 deficient for binding to Nmd4 is impaired for its NMD role. A *upf1Δ* strain was transformed with plasmids expressing N-terminal tagged full-length Upf1 (pUpf1), to restore NMD, a truncated WT version of the protein lacking the CH domain (UPF1-HD-Ct), which can partially complement NMD, or UPF1-HD-Ct single point mutants (G243R or G377R) that are unable to interact with Nmd4. A double mutant *upf1Δ/nmd4Δ* strain shows an exacerbated NMD defect phenotype, which is not rescued by the expression of UPF1-HD-Ct single point mutants (G243R or G377R) that are unable to interact with Nmd4. Two natural NMD substrates, the uORF containing DAL7 transcript (left) and the unspliced pre-mRNA for the ribosomal protein Rpl28 (right) were used to estimate NMD efficiency and their levels were measured by RT-qPCR. The obtained values were normalized for total RNA amounts using PGK1 and spliced RPL28 transcripts and all the results were adjusted to the situation in which full-length Upf1 was expressed (marked "pUpf1"). The experiments were repeated at least 3 times and the individual results are indicated as dots. Error bars represent standard deviation. The *p*-values of *t* tests (Welch variant with continuity correction) for comparisons between results are indicated. The data underlying this figure can be found in S3 Data. (F) Pull-down experiment showing that the His$_6$-ZZ-Nmd4 single point mutants (R210E or W216A) or the double point mutant (M2; R210E/W216A) affect the binding to Upf1. Input and eluate from NiNTA magnetic beads samples were analyzed on 12% SDS/PAGE and Coomassie blue staining. As control, Upf1-HD was incubated with the His$_6$-ZZ tag. Three biologically independent experiments were performed and a representative image is shown. (G) The Nmd4 mutants are less potent activators of Upf1-HD ATPase activity than WT Nmd4. The assay was similar to that presented in Fig 2B. The data underlying this figure can be found in S4 Data. HD, helicase domain; NMD, nonsense-mediated mRNA decay; uORF, upstream open reading frame.

*cerevisiae* Nmd4 and Upf1-HD proteins according to our crystal structure and our site-directed mutagenesis experiments (Figs 1C, 1E, 3A and 3B). This prompted us to investigate whether metazoan SMG6 proteins might also contain an arm-like region in their sequence. We focused our attention on the low complexity region, encompassing residues to 207 to 580, which had been identified as being involved in the phosphorylation-independent interaction between human SMG6 and UPF1 proteins [31]. Interestingly, a $^{448}$RGX$_5$LWDP$^{458}$ motif (numbering based on human SMG6 protein) reminiscent of the fungal $^{210}$RGX$_{3-4}$LWXP$^{218}$ motif (numbering based on *S. cerevisiae* Nmd4 protein) is present in metazoan SMG6 proteins (Fig 4A). Based on this observation, we generated a model of the complex between human UPF1-HD and the region 398 to 494 of SMG6 using AlphaFold3 software (S9A–S9D Fig; [56]). In this model, the SMG6 fragment binds to the same region of UPF1-HD as the Nmd4 « arm » (S9E Fig). In particular, the R448 and W456 side chains of SMG6 match almost perfectly with R210 and W216 side chains of *S. cerevisiae* Nmd4 (S9E Fig), suggesting that this conserved region from SMG6 is involved in the interaction between the SMG6 and UPF1-HD proteins. To validate this hypothesis, we ectopically expressed the region comprising residues 207 to 580 of human SMG6 fused to a C-terminal HA tag (SMG6-[207–580]-HA) and human UPF1-HD (residues 295 to 921 fused to a C-terminal Flag-tag; UPF1-HD-Flag) in human HEK293T cells, as these regions have previously been shown to be responsible for the phosphorylation-independent interaction between these 2 proteins [31,36]. Compared to full-length UPF1 and SMG6 proteins, these constructs also preclude our findings from any interference due to the phosphorylation-dependent interaction occurring between the C-terminus of UPF1 and the 14-3-3 domain of SMG6 [31]. Co-immunoprecipitation (co-IP) experiments of UPF1-HD-Flag revealed a specific interaction with wild-type SMG6-[207–580]-HA (Fig 4B). We then generated SMG6 point mutants R448E and W456A, equivalent to *S. cerevisiae* Nmd4 mutants R210E and W216A, respectively, as well as the R448E/W456A double mutant (M2). As expected, the single or double point SMG6 mutants showed weaker interactions with UPF1-HD than wild-type SMG6 (Fig 4B). Therefore, these 2 SMG6 residues (R448 and W456) are very important for the interaction with UPF1-HD, suggesting that the SMG6 « arm-like » motif is the critical region involved in the previously described phospho-independent interaction between SMG6 and the UPF1 helicase domain [31,36].

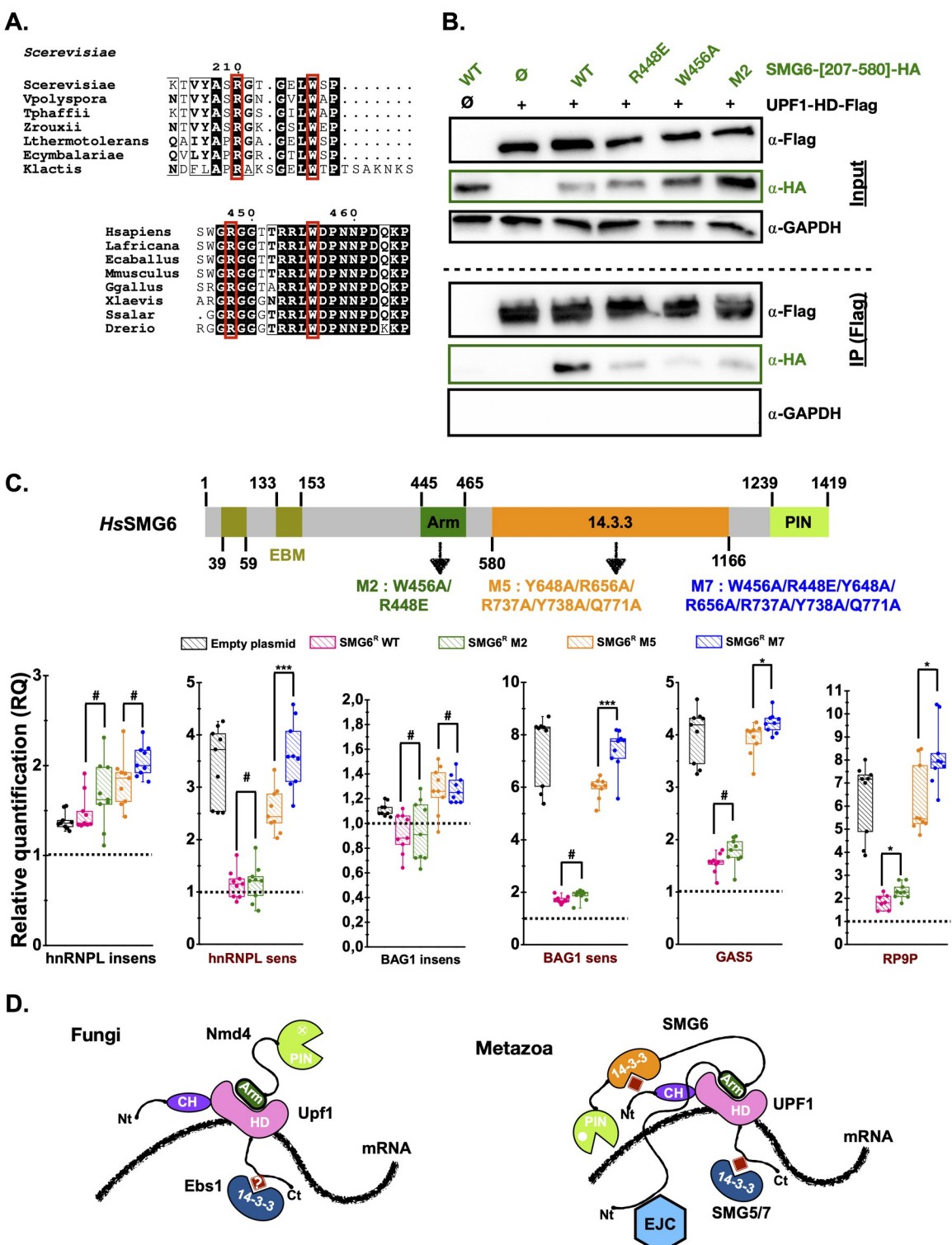

**Fig 4. An evolutionary conserved « arm » in metazoan SMG6 is critical for the interaction with UPF1-HD and influences the levels of NMD substrates.** (A) Zoom-in on the sequence alignments of the C-terminal end of fungal Nmd4 proteins and of the metazoan SMG6 region encompassing residues 445–465 from human SMG6. Strictly conserved residues are in white on a black background. Strongly conserved residues are in bold. Residues mutated in this study are highlighted by red boxes. This figure was generated using the ENDscript server [76]. (B) The SMG6 R448 and W456 residues corresponding respectively to R210 and W216 from Nmd4 are important for the phospho-independent interaction between human UPF1-HD and SMG6-[207–580] domain in cellulo. Co-immunoprecipitation

experiments showing the reduced interaction of the 3 human SMG6-[207–580] point mutants (R448E, W456A, M2) with human UPF1-HD (residues 295–914) as compared to human SMG6-[207–580] WT. The human UPF1-HD-3xFlag and the different human SMG6-[207–580]-HA variants were expressed in HEK293T cells and purified using anti-Flag beads. Co-immunoprecipitated proteins were detected by western blot using the indicated antibodies. Co-IP experiments were performed in the presence of benzonase to exclude a potential role of nucleic acids in these interactions. Three biologically independent experiments were performed and a representative image is shown. (C) The phospho-independent SMG6-UPF1 interaction contributes to NMD in human cells. Top: Schematic representation of human SMG6 protein with domain boundaries indicated. EBM: EJC-binding motif. 14.3.3 (also known as TPR for Tetratrico Peptide Repeat) is a domain involved in the recognition of phosphorylated Ser or Thr. The different protein domains are shown with a color code. Mutants investigated in this study are indicated below the diagram. Bottom: Box plot representation of the effect of SMG6 mutations on the stability of endogenous NMD substrates. Human cells were depleted for endogenous SMG6 using shRNAs and transformed using either empty plasmid, or various SMG6 constructs resistant to the shRNAs. Four previously described endogenous NMD substrates (highlighted in dark red) and 2 NMD insensitive transcripts (labeled in black) were quantified using RT-qPCR. The RQ value of 1 observed for each transcript in parental cell lines is depicted by an horizontal black dashed line. Statistical differences were determined from 9 data points (3 technical replicates for each of the 3 biological replicates) using an independent two-tailed Student's $t$ test to compare RQ values for the « arm » mutants (M2 and M7) with those of the corresponding controls (WT and M5, respectively). *** $p < 0.001$; * $0.01 < p < 0.05$; #: not statistically significant. The data underlying this figure can be found in S5 Data. (D) Proposed models for the interaction of Upf1/UPF1 with Nmd4/SMG6 and Ebs1/SMG5-7 proteins in fungi (left) and metazoa (right). Phosphorylated Ser/Thr are depicted as red squares. Putative phosphorylation sites in yeast Upf1 are indicated with a question mark (?). EJC, exon junction complex; NMD, nonsense-mediated mRNA decay; RQ, relative quantification.

Next, we tested whether the M2 mutant was affected in its ability to trigger the degradation of endogenous NMD substrates. We therefore depleted human HeLa cells of endogenous SMG6 protein by shRNAs and expressed shRNA-resistant versions of WT or M2 SMG6 proteins (SMG6[R]) in these cell lines (S10 Fig). We then measured the levels of 4 previously described endogenous NMD substrates (hnRNPL sensitive, BAG1 sensitive, GAS5 and RP9P; [44,57]) as well as 2 endogenous NMD-insensitive transcripts (hnRNPL and BAG1 insensitive) by RT-qPCR. As expected, the transfection of the empty plasmid into HeLa cells depleted in endogenous SMG6 protein resulted in the specific stabilization of the NMD-sensitive substrates (relative quantification (RQ) values >> 1) but not of hnRNPL and BAG1 insensitive isoforms (RQ values close to 1; Fig 4C). Complementation with SMG6[R] WT significantly reduced the levels of NMD-sensitive mRNAs, to levels close to those observed in parental cell lines, validating our experimental approach. At the same time, the expression of the SMG6[R] M2 mutant did not significantly increase the levels of the endogenous NMD substrates, with the exception of RP9P, compared to cells expressing the RNAi-resistant version of WT SMG6 (Fig 4C). The phosphorylation-dependent interaction between UPF1 and SMG6 could compensate for the disruption of the phosphorylation-independent interaction by maintaining the formation of the UPF1-SMG6 complex. We therefore generated the M5 mutant carrying 5 mutations in the 14-3-3 domain of SMG6 (Fig 4C), as these mutations or related ones have previously been shown to strongly reduce the interaction between SMG6 and UPF1 full-length proteins [36,37]. In parallel, we constructed the M7 mutant corresponding to the combination of the M2 and M5 mutations (Fig 4C). The M5 mutant strongly increased the levels of NMD-sensitive but not NMD-insensitive transcripts, in agreement with previous results obtained with a related SMG6 14-3-3 mutant [37]. As expected, the M7 mutant led to an increased stabilization in the same transcripts as the M5 mutant, comparable to those observed when the depletion of endogenous SMG6 was not compensated for by the expression of RNAi-resistant forms of SMG6. This indicates that the interaction between the helicase domain of UPF1 and the « arm-like » motif of SMG6 is functionally important in a context where the so-called phospho-dependent interaction between UPF1 and SMG6 is already affected. This confirms the synergistic role of SMG6's "arm-like" motif and 14-3-3 domain for optimal degradation of NMD substrates. In this model, the 14-3-3 domain plays a predominant role, in line with the previously described preferential binding of SMG6 to phosphorylated UPF1 [37], while the « arm-like » domain plays a secondary role. This observation is reminiscent of the importance of

the Nmd4 protein in the yeast NMD pathway, which can only be detected in a yeast strain expressing a truncated version of the protein Upf1 (Upf1-HD-Ct; Fig 3E and [45]). The effects on NMD observed following the perturbation of the interaction between the « arm » of SMG6 or Nmd4 proteins with the corresponding Upf1 protein appear weak under laboratory conditions. However, the conservation of this newly identified binding motif in metazoan SMG6 or fungal Nmd4 proteins, underlines its importance. Whether this is through its impact on the enzymatic properties of the Upf1 RNA helicase, as demonstrated here for the yeast proteins, remains to be clarified.

## Discussion

Despite decades of studies performed in various eukaryotic organisms, the molecular mechanisms responsible for the rapid degradation of PTC-containing mRNAs by the NMD pathway still remain elusive. Several recent publications support the existence of a common mechanism conserved from yeast to human, which may be more elaborate in metazoans to cope with the greater complexity of their mRNA maturation and degradation processes. Our results support a universal model of NMD in which Upf1's fungal partners, namely Nmd4 and Ebs1 proteins [45,46], would play roles similar to those of the metazoan SMG5-6-7 proteins (Fig 4D). The presence of a 14-3-3 domain in both yeast Ebs1 and metazoan SMG5-6-7 proteins suggests that Ebs1 may interact with a phosphorylated form of Upf1, as do human SMG5-6-7 proteins [37,51,58]. Since Ebs1 binds preferentially to the C-terminal extension of Upf1 [45], it could correspond to the SMG5-SMG7 heterodimer, which also interacts preferentially with Upf1 C-ter [37]. It is not known whether this interaction is phospho-dependent, but this possibility cannot be ruled out as Upf1 from *S. cerevisiae* has been shown to be phosphorylated [59,60] and to co-purify with the Hrr25 protein kinase [45]. Here, we focused on Nmd4, revealing a bipartite protein with a PIN domain similar to those found in metazoan SMG6 and SMG5 and an « arm » region, which is essential for the interaction between Upf1-HD and Nmd4 (Figs 2A and S5). Nmd4 stimulates Upf1-HD ATPase activity and strengthens Upf1-HD binding to RNA (Figs 2B and S6 and Table 1). ATP hydrolysis by Upf1 is essential for the NMD mechanism [61–63] and this could explain why the interaction between Upf1 and the « arm » domain of Nmd4 is critical for mRNA degradation under conditions where NMD is already partially compromised (Fig 3E).

Interestingly, we show that a motif equivalent to the yeast Nmd4 « arm » is also present and important in SMG6 (but absent in SMG5), one of the central factors of NMD in metazoans, and which is responsible not only for the recruitment of UPF1 but also for the endonucleolytic cleavage of NMD substrates [32,33]. The recruitment of the isolated SMG6 PIN endonuclease domain to NMD substrates is not sufficient to trigger their elimination, as demonstrated by tethering assays [36], indicating that additional interactions between SMG6 and other partners (UPF1, EJC…) may be important to initiate mRNA decay [35]. The phospho-dependent interaction between SMG6's 14-3-3 domain and UPF1 could be one such key interaction, since the M5 variant of SMG6's 14-3-3 domain strongly affects NMD (Fig 4C). However, the tethering of the same mutant destabilizes an NMD reporter in human cells [36], indicating that other SMG6 regions, such as the EJC-binding motif [35,36], may be important. Here, we show that the arm-like region identified in metazoan SMG6 proteins is also involved in the optimal elimination of human endogenous NMD substrates, particularly when NMD efficiency is already affected (Fig 4C), similar to our observations in yeast. This is confirmed by the inability of a UPF1 protein lacking the stalk region, which is involved in the Nmd4-Upf1 interaction in yeast, to restore RNA decay in SMG6 tethering assays performed in UPF1 KD cells [36]. SMG6 clearly emerges as a protein with multiple binding sites for UPF1 since while this article

was under revision, a short linear motif (amino acids 406 to 413) located just upstream of the « arm-like » motif (amino acids 445 to 465) identified in our study, was shown to interact with the CH domain from human UPF1 (Fig 4D; [64]). This motif competes with UPF2 to interact with UPF1 CH, further supporting the evolutionary conserved existence of 2 NMD complexes assembled around UPF1, namely the UPF1-23 and the UPF1-effector complex as previously shown in yeast [45]. Altogether, this supports that the conserved phosphorylation-independent binding mode between Upf1 and Nmd4 in yeasts and UPF1/SMG6 in metazoa might be more important for NMD than the nuclease activity since the catalytic residues are absent in the PIN domain from fungal Nmd4 proteins.

Our study illustrates the existence of a complex interaction network between various NMD factors for the optimal degradation of aberrant mRNAs, which has been rewired and extended with the emergence of new components from unicellular to multicellular eukaryotes. This is reminiscent of what has been described for the eukaryotic mRNA decapping network, where many interactions are achieved through linear motifs that can pass from one protein to another in different organisms while maintaining the interaction networks [65,66]. This makes detailed analysis of the effect of a specific binding site in multivalent proteins such as SMG6 and UPF1 particularly difficult, and could explain the limited effect of the mutations of the « arm » region of human SMG6 on NMD substrate degradation.

In conclusion, our crystal structure of the complex formed between the helicase domain of Upf1 and the *S. cerevisiae* Nmd4 protein has enabled us to identify a previously uncharacterized but conserved region within metazoan SMG6 proteins. We have also shown that equivalent regions of yeast Nmd4 and human SMG6 proteins are important for the degradation of NMD substrates under conditions where the NMD pathway is already affected. This suggests that, during evolution, a subtle mechanism has been preserved to ensure the correct decay of aberrant mRNAs containing premature stop codons or transcripts with long 3′ UTRs. Further studies in different model systems such as yeast and human cells will be needed to understand the interplay between the different NMD factors, but there is increasing evidence for the existence of a single, conserved mechanism for the NMD pathway in eukaryotes, around which species-specific features have evolved.

## Materials and methods

### Cloning

The *S. cerevisiae* Nmd4 coding sequence was amplified by PCR with oligonucleotides oMG504 and oMG505 and inserted using *BamH*I and *Not*I enzymes into a homemade pET28b-His$_6$-ZZ plasmid (kind gift from Dr. D. Hazra), yielding pMG838 (details about oligonucleotides and plasmids are listed in S3 and S4 Tables). The ZZ-tag consists in a tandem of the Z-domain from *Staphylococcus aureus* protein A and was used as an enhancer of protein expression and stability. The Nmd4 point mutants were generated by one-step site-directed mutagenesis according to Zheng and colleagues [67]. The pMG1003 plasmid expressing the Nmd4 PIN domain (residues 1 to 167) was generated by introducing a stop codon by one-step site-directed mutagenesis using oligonucleotides oMG726/727. To generate the pMG1044 plasmid expressing Nmd4 arm (residues 168 to 218), the DNA region encoding for this region was amplified using oMG730 and oMG505 primers and inserted pET28b-His$_6$-ZZ plasmid as described above. The pHL1670 plasmid expressing CBP-His$_6$-ZZ-ScNmd4 was generated by inserting an in vitro synthesized DNA fragment (obtained from Integrated DNA Technology) encoding for CBP in the pMG838 plasmid using NcoI and NdeI, restrictions enzymes.

The coding sequence for residues 221 to 851 from *S. cerevisiae* Upf1 (hereafter named Upf1-HD) was amplified by PCR and inserted into various vectors dedicated to protein

overexpression in *E. coli* (see S3 and S4 Tables for details about oligonucleotides and plasmids). The different Upf1-HD point mutants were generated as described above for Nmd4 mutants.

## Expression and purification of recombinant proteins

The Nmd4 and Upf1-HD proteins were overexpressed in *E. coli* BL21 (DE3) Gold cells in 1 L of auto-inducible Terrific broth (TBAI; ForMedium; #AIMTB0260) containing either ampicillin (100 μg/mL for plasmids encoding Upf1-HD) or kanamycin (100 μg/mL for plasmids encoding Nmd4 fragments). After a 3 h incubation at 37°C, the bacteria were incubated overnight at 25°C. Cells were harvested by centrifugation at 4,000 rcf for 30 min at 4°C, resuspended in buffer $A_{200}$ (20 mM Tris-HCl (pH 7.5), 200 mM NaCl, 5 mM 2-mercaptoethanol, 5 mM $MgCl_2$, where the number in lower script corresponds to the NaCl concentration in mM) and lysed by sonication after addition of phenylmethylsulfonyl fluoride (PMSF; 500 μM).

The *S. cerevisiae* $His_6$-ZZ-Nmd4 wild-type protein (full-length or PIN domain) and mutants thereof, were first purified on a Protino Ni-NTA agarose resin pre-equilibrated with buffer $A_{200}$. After successive washing steps using buffers $A_{200}$ and then $A_{200}$ supplemented with 20 mM imidazole pH 7, the proteins were eluted using buffer $A_{200}$ supplemented with 400 mM imidazole pH 7. When necessary the $His_6$-ZZ tag was removed upon overnight incubation with 200 μg of rhinovirus 3C protease (bearing $His_6$ and GST tags) under dialysis condition against buffer $B_{200}$ (100 mM NaCitrate (pH 5.6), 200 mM NaCl, 5 mM $MgCl_2$, 5 mM 2-mercaptoethanol). After a centrifugation step to pellet precipitated proteins, the samples were loaded on a HiTrap Heparin (Cytiva) column pre-equilibrated in buffer $B_{50}$ and eluted using a linear gradient from 100% $B_{50}$ to 100% $B_{1,000}$. The proteins of interest were next purified on a S75-16/60 size-exclusion column (Cytiva) equilibrated with buffer $B_{200}$. The CBP-$His_6$-ZZ-Nmd4 protein was purified using the same protocol as the $His_6$-ZZ-Nmd4 proteins, with no 3C cleavage step.

The Upf1-HD-$His_6$ wild-type protein was first purified on a TALON resin (Clontech) equilibrated with buffer $A_{200}$. After successive washing steps using buffer $A_{200}$, $A_{1,000}$, $A_{200}$, and $A_{200}$ supplemented with 20 mM imidazole (pH 7), respectively, the protein was eluted using buffer $A_{200}$ supplemented with 400 mM imidazole (pH 7). The purification was pursued by a HiTrap Heparin (Cytiva) column using buffers $A_{50}$ and $A_{1,000}$ and then by a S200-16/60 size-exclusion column (Cytiva) equilibrated with buffer $A_{200}$.

To purify untagged Upf1-HD wild-type and mutants thereof, the bacterial pellets were resuspended in buffer $C_{200}$ (20 mM Hepes (pH 7.5), 200 mM NaCl, 5 mM $MgCl_2$, 5 mM 2-mercaptoethanol) and lysed by sonication after addition of PMSF. The CBP-$His_6$-ZZ-Upf1-HD proteins were first purified on a Protino Ni-NTA agarose resin (Macherey-Nagel). After successive washing steps using buffers $C_{200}$ first, $C_{1,000}$ second, $C_{200}$ third, and finally $C_{200}$ supplemented with 20 mM imidazole (pH 7), the proteins were eluted using buffer $C_{200}$ supplemented with 400 mM imidazole (pH 7). The CBP-$His_6$-ZZ tag was next removed upon overnight incubation with 3C protease under dialysis condition against buffer $C_{100}$. The CBP-$His_6$-ZZ tag was removed by incubation with Ni-NTA resin. The untagged Upf1-HD proteins were then loaded on a HiTrapS (Cytiva) column pre-equilibrated in buffer $C_{50}$ and eluted using a linear gradient from 100% $C_{50}$ to 100% $C_{1,000}$. The proteins of interest were next purified on a S200-16/60 size-exclusion column (Cytiva) equilibrated with buffer $C_{200}$.

## Structure of the Nmd4 PIN domain

The Nmd4 PIN domain (residues 1 to 167) was expressed from pMG1003 plasmid and purified as described above. The best crystals were obtained at 4°C by mixing an equal volume of Nmd4 PIN domain (10 mg/mL) and of crystallization solution (1M Na citrate, 0.1 M Na

cacodylate (pH 6.5), 0.2 M lithium acetate). Crystals were cryo-protected by transfer into crystallization solution supplemented with first 15% and then 30% glycerol and flash-cooled in liquid nitrogen. Data were collected at 100 K on the PROXIMA-2A beamline at Synchrotron SOLEIL, Saint-Aubin, France [68]. One data set was collected from a single crystal and was processed using XDS, merged, scaled using XSCALE [69]. Statistics for data processing are summarized in S1 Table. The structure was solved by molecular replacement with the PHASER program [70] using the structure of the PIN domain of *Kluyveromyces lactis* Nmd4 (*Kl*Nmd4; PDB code: 7QHY; [49]). Several cycles of building and refinement were performed using COOT (57) and BUSTER (58), respectively. The final model has R and $R_{free}$ factors of 18.2% and 19.2%, respectively at 1.8 Å resolution (see S1 Table for refinement statistics).

## Determination of the structure of the Nmd4/Upf1-HD complex

To reconstitute the Nmd4/Upf1-HD complex, the proteins were purified separately as described above for the TALON resin and ion-exchange chromatography steps with the exception that the $His_6$-ZZ tag was cleaved by incubating eluted $His_6$-ZZ-Nmd4 protein for 1 h at room temperature with 3C protease (200 µg). After removal of the $His_6$-ZZ tag and tagged 3C protease upon incubation on TALON resin, the Nmd4 protein was incubated with Upf1-HD in a 2:1 molar ratio for 1 h at 4°C. The mix was then injected on a S200-16/60 size-exclusion column (Cytiva) equilibrated with buffer $B_{200}$. The fractions corresponding to the peak containing the Nmd4/Upf1-HD complex were pooled and the complex was concentrated up to 6.7 mg/mL for crystallization trials. Initial crystals were obtained at 4°C by mixing 150 nL of the Nmd4/Upf1-HD complex (6.7 mg/mL) incubated with a 10-fold molar excess of the ATPɣS, a non-hydrolyzable ATP analog, with an equal volume of crystallization solution (35% Tacsimate pH 7). Larger crystals were obtained by increasing volumes from 150 nL to 1 µL and by adding 100 or 200 µL of paraffin oil onto the crystallization week to slow-down equilibration. Crystals were cryo-protected by transfer into crystallization solution supplemented with first 15% and then 30% ethylene glycol and flash-cooled in liquid nitrogen. Data were collected at 100 K on the PROXIMA-2A beamline at Synchrotron SOLEIL, Saint-Aubin, France [68]. Three data sets collected from 3 isomorphous crystals were processed using XDS, merged, scaled using XSCALE [69] and analyzed using the STARANISO server as diffraction was highly anisotropic [71]. Crystals belong to space group I222 with one copy of the Nmd4/Upf1-HD complex in the asymmetric unit. Statistics for data processing are summarized in S1 Table. The structure of the complex was solved by molecular replacement with the PHASER program [70] using the previously solved crystal structure of *S. cerevisiae* Upf1 (PDB code: 2XZL; [55]) and of *S. cerevisiae* Nmd4 PIN domain solved in this study. For molecular replacement, the RecA1, RecA2, 1B, 1C, and stalk domains from Upf1 were searched separately. Several cycles of building and refinement were performed using COOT (57) and BUSTER (58), respectively. The high-resolution structure of the PIN domain (residues 1 to 167), which we solved concomitantly, was used to build the model of this domain (rmsd of 0.5 Å over 163 Cα atoms between both structures). The final model has R and $R_{free}$ factors of 20.2% and 22.1%, respectively, at 2.4 Å resolution (see S1 Table for refinement statistics). The final model encompasses residues 224 to 851 from Upf1-HD and residues 1 to 216 from Nmd4.

## Generation of the human UPF1-HD/SMG6 model

Five models of the 3D structure of human UPF1-HD (region 295 to 925) bound to SMG6 region 398 to 494 (i.e., containing the conserved region 445 to 465) were generated using the recently launched version of AlphaFold program (version 3; AF3), with the seed auto option as defined in the AlphaFold server (https://alphafoldserver.com/; [56]).

## Pull-down experiments

For pull-down experiments, all the $His_6$-ZZ-Nmd4 or CBP-$His_6$-ZZ-Nmd4 proteins were purified as described above except that the 3C digestion step was omitted. The untagged Upf1-HD proteins were purified as described above. All proteins were free of nucleic acids according to the $OD_{280nm}/OD_{260nm}$ ratio.

His-pull-down experiments were performed by mixing 0.5 nmol of $His_6$-ZZ-Nmd4 proteins with 0.5 nmol of the various Upf1-HD proteins. Binding buffer (20 mM Tris-HCl (pH 7.5), 150 mM NaCl, 50 mM imidazole (pH 7), 10% glycerol) was added to a final volume of 60 μL. The reaction mixtures were incubated on ice for 1 h, and 10 μL were withdrawn as « input » fraction. The remaining 50 μL were incubated with 40 μL of HisPur Ni-NTA magnetic beads (Thermo) pre-equilibrated with binding buffer in a final volume of 200 μL at 4°C for 1 h on a rotating wheel. Beads were washed 3 times with 500 μL of binding buffer. Bound proteins were eluted with 50 μL of elution buffer (binding buffer supplemented with 200 mM imidazole (pH 7)), and 4 μL of « input » and 20 μL of « elution » fractions were resolved on SDS-PAGE and visualized by Coomassie blue staining.

For CBP pull-down experiments, the Calmodulin Sepharose 4B (Cytiva) beads were washed twice in blocking buffer (20 mM Hepes-NaOH (pH 7.5), 150 mM NaCl, 0.1% NP-40), incubated 2 h with 4.5 μg/mL glycogen, 0.05 mg/ml yeast tRNAs, 0.5 mg/ml BSA. Beads were washed 3 times with blocking buffer and resuspended in binding buffer (20 mM Hepes-NaOH (pH 7.5), 125 mM NaCl, 2 mM magnesium acetate, 2 mM imidazole (pH 7), 2 mM $CaCl_2$, 0.1% NP-40, 10% glycerol, 2 mM DTT). Pull-down experiments were performed by mixing 0.5 nmol of CBP-$His_6$-ZZ or CBP-$His_6$-ZZ-Nmd4 proteins with 0.5 nmol of the various Upf1-HD proteins. Binding buffer was added to a final volume of 60 μL. The reaction mixtures were incubated at 22°C for 0.5 h, and 10 μL were withdrawn as « input » fraction. The remaining 50 μL were incubated with 25 μL of beads in a final volume of 200 μL at 4°C for 1 h on a rotating wheel. Beads were washed 3 times with 800 μL of binding buffer. Bound proteins were eluted with 25 μL of 5× loading buffer, and 5 μL of « input » and 15 μL of « elution » fractions were resolved on SDS-PAGE and visualized by Coomassie blue staining.

## Isothermal titration calorimetry (ITC)

For ITC experiments, the various Upf1-HD-$His_6$ and $His_6$-ZZ-Nmd4 proteins (wild-type and truncated forms) were purified as described above except that the last size exclusion chromatography step was performed with buffer $D_{200}$ (100 mM Bis-Tris (pH 6.5), 200 mM NaCl, 5 mM $MgCl_2$, 5 mM 2-mercaptoethanol).

Measurements were performed at 20°C and 500 rpm stirring using an ITC-200 microcalorimeter (MicroCal). For the titration of Upf1-HD-$His_6$ (30 to 35 μM) with $His_6$-ZZ-Nmd4 proteins (260 μM for full-length Nmd4 and « arm », 330 μM for Nmd4 PIN) or $His_6$-ZZ (277 μM; control), 20 injections of 2 μL of $His_6$-ZZ-Nmd4 were added to Upf1-HD-$His_6$ at intervals of 180 s. The heat of dilution of the titrant was determined from the peaks measured after full saturation of the target by the titrant. A theoretical curve assuming a one-binding site model calculated with the ORIGIN software gave the best fit to the experimental data. This software uses the relationship between the heat generated by each injection and ΔH (enthalpy change), Ka (association binding constant), n (the number of binding site per monomer), total Upf1-HD-$His_6$ concentration, and the free and total $His_6$-ZZ-Nmd4 or $His_6$-ZZ concentrations. For RNA-binding experiments, Upf1-HD-$His_6$ (20 to 22 μM), $His_6$-ZZ-Nmd4 (30 μM), or the Nmd4/Upf1-HD complex (28 μM) were titrated with RNA poly(U)$_{30}$ (230 to 280 μM; Dharmacon, Horizon Discovery, Cambridge, United Kingdom) in the following buffer: 100

mM Bis-Tris (pH 6.5), 133 mM NaCl, 5 mM MgCl$_2$, 5 mM 2-mercaptoethanol using the same protocol described above.

## ATPase assays

Proteins purified in buffer D$_{200}$ (100 mM BisTris (pH 6.5), 200 mM NaCl, 5 mM 2-mercaptoethanol, 5 mM MgCl$_2$) were thawed on ice and diluted in ATPase buffer 1× (50 mM MES (pH 6.5); 50 mM K Acetate; 5 mM MgAc$_2$; 2 mM DTT; 0.1 mg/ml BSA). Upf1-HD (1.25 pmol) was mixed with its partner (25 pmol) in a final volume of 8 μL and incubated for 1 h at 4˚C. For each reaction, 12 μL of ATP premix (0.1 μL of ATPγ$^{32}$P (6,000 Ci/mmol; Perkin Elmer); ATPase Buffer 1×; 2 mM ATP, 0.4 mg/ml RNA poly(U)) were added and the reaction vials were sequentially placed at 30˚C every 30 s. For each time point, 4 μL were quenched in 400 μL of Charcoal mix (10% activated charcoal (Sigma; #C5510); 10 mM EDTA (pH 8.0)), vortexed, and kept on ice. At the end of the time course, quenched reactions were centrifuged 15 min at 14,000 rpm and 4˚C, and 140 μL of supernatant were transferred to fresh tubes and scintillation was counted using the Cerenkov method.

## In vivo studies in *Saccharomyces cerevisiae*

**Yeast strains.** C-terminal tagging of Nmd4 and of Nmd4-[1–168] was done by homologous recombination in BY4741 and LMA2194 (Upf1-TAP, [72]) using PCR fragments that amplified a 3HA-KanMX cassette [73] with 50 nt ends targeting its insertion downstream the last codon of Nmd4, or downstream codon 168 of Nmd4 for the truncated version. Oligonucleotides used were CS1543 and CS1544 for tagging full-length Nmd4 and CS1542 and CS1544 for tagging the truncated version of Nmd4 (S3 and S5 Tables). Insertion was tested by PCR on genomic DNA and by immunoblot against the introduced HA tag. A similar strategy was used to build the strain LMA4490 (*upf1Δ*, Nmd4-HA) starting from an Nmd4-HA strain.

**Plasmid generation.** Vectors for the expression of N-terminal tagged UPF1 full length and UPF1[208–971] variants were built using Gibson assembly using plasmid p1233 (pCM189-NTAP) digested with *Not*I as destination vector. For G243R, 2 PCR products were obtained using as template a plasmid with the coding sequence of yeast UPF1 using the oligonucleotides CS1359, CS1664, CS1663, and CS1364. For G377R, the pairs of oligonucleotides were CS1359 and CS1666 and CS1665 and CS1364. The obtained plasmids, p1760 (G243R) and p1761 (G377R) were verified by sequencing. Next, the region corresponding to the 208–971 amino acids of UPF1 was amplified from these plasmids using oligonucleotides CS1362 and CS1364 and cloned in *Not*I digested p1233 to obtain p1764 and p1765 plasmids, also verified by sequencing.

**Immunoblots.** HA-tagged proteins were detected using anti-HA-peroxidase and TAP-tagged proteins were detected using peroxidase-anti-peroxidase complexes.

**Purification of TAP-tagged proteins.** Purification of tagged proteins was performed as described in Namane and Saveanu [74]. Briefly, cells from 2 liters of yeast culture in exponential growth phase were recovered by centrifugation, washed with cold water and suspended in a breaking buffer containing 20 mM HEPES, 100 mM potassium acetate, 5 mM magnesium chloride, and protease inhibitors (Roche). A FastPrep-24 (MP Bio) system was used for mechanical lysis of yeast cells with glass beads; the lysate was cleared by centrifugation and incubated with home-made IgG-magnetic beads for 1 h at 4˚C. Beads were washed with a buffer containing 20 mM HEPES (pH 7.4), and proteins were eluted either by direct denaturation in a 10 mM Tris-HCl, 1 mM EDTA, 2% SDS, pH 8 buffer for 10 min at 65˚C or by incubation for 10 min at room temperature with a buffer containing 2 M MgCl$_2$. Elution with

$MgCl_2$ was followed with protein precipitation with methanol-chloroform or 8% TCA in the presence of sodium deoxycholate.

**Yeast RNA extraction and RT-qPCR analysis.** RNA measurements were done using hot phenol extraction from cells growing exponentially, followed by DNase I treatment and reverse transcription with oligonucleotides specific for the tested transcripts. Quantitative PCR reactions with SybrGreen detection were performed using the SsoAdvanced Universal SYBR Green Supermix (Bio-Rad Cat# 1725270) and the following pairs of oligonucleotides: CS1429 and CS1430 for DAL7, CS1020 and CS1021 for PGK1, CS887 and CS888 for the unspliced RPL28 transcript, and CS889 and CS946 for the mature RPL28 mRNA. The amplification was done in a Bio-Rad CFX96 machine, with a denaturing initial step, 95˚C for 3 min, followed by 40 cycles of 95˚C for 10 s and 60˚C for 30 s.

## Human cell experiments

**Cloning.** The coding sequences for human UPF1-HD (residues 295 to 914) and SMG6-[207–580] WT regions were amplified using oligonucleotides listed in S3 Table and inserted using *Nhe*I and *Sma*I enzymes into pCI-Neo plasmids containing the sequences coding for C-terminal tags (HA for UPF1 and 3xFlag for SMG6), yielding pMG1078 and pMG1081 (S4 Table), respectively. The different point mutants were generated by one-step site-directed mutagenesis according to Zheng and colleagues [67]. Details about primers and plasmids used in this study are summarized in S3 and S4 Tables.

The pSUPuro plasmid to knockdown SMG6 gene by expressing a shRNA directed against SMG6 transcripts and the plasmid to express an RNAi resistant version of WT SMG6 (p700) were kindly shared by Pr. Oliver Mühlemann (Univ. Bern; Switzerland; S4 Table). The pMG1125 plasmid expressing an RNAi resistant version of the SMG6 M2 (R448E/W456A) mutant was generated by a 2 steps site-directed mutagenesis protocol using oMG795/796 to generate the R448E mutant and then oMG797/798 to generate the R448E/W456A double mutant. To generate the plasmids encoding for SMG6 proteins harboring the 5 mutations in the 14-3-3 domain (M5 and M7), an in vitro synthesized DNA fragment with the corresponding mutations (obtained from Integrated DNA Technologies, Belgium) was digested using *Kpn2*I and *Bsp1407*I restriction enzymes and then inserted into p700 and pMG1125 plasmids digested with the same enzymes to generate plasmids pMG1136 (mutant M5) and pMG1137 (mutant M7), respectively.

**Co-immunoprecipitation and western blotting.** $4.1 \times 10^6$ HEK293T cells were seeded in 10-cm plates and cultured in DMEM complete medium (DMEM medium (Gibco) supplemented with 10% fetal bovine serum (Gibco), 200 U/mL of penicillin and 200 µg/mL streptomycin) at 37˚C and 5% of $CO_2$. On the next day, cells were transfected with 1 µg and 4 µg of plasmids encoding UPF1-[295–914]-3×Flag or SMG6-[207–580]-HA, respectively, using Lipofectamine 2000 (Thermo Fisher). For control experiments, 4 or 1 µg of empty plasmids were mixed with 1 µg and 4 µg of plasmids encoding for UPF1-[295–914]-3×Flag or SMG6-[207–580]-HA, respectively. After 24 h, cells were washed with 1xPBS (Gibco) and harvested in lysis buffer (50 mM Tris-HCl (pH 8), 150 mM NaCl, 0.1% NP-40, 1 mM EDTA, 10% glycerol and cOmplete EDTA-free Protease inhibitor (1×PIC)). Lysates were rotated at 4˚C for 30 min and then centrifuged at 16,000×g for 20 min at 4˚C. Cell extracts were incubated overnight at 4˚C with magnetic anti-Flag agarose beads (Pierce; #A36797) pre-equilibrated with lysis buffer supplemented with benzonase (1 mg). After several washing steps with the lysis buffer, the beads were mixed with 1.5× loading buffer and then loaded on a 4% to 15% pre-casted protein gels (BioRad; #456–1084) prior to transfer on PVDF membrane for western blotting using anti-Flag, anti-HA, or anti-GAPDH (loading control) antibodies (S6 Table). HRP-coupled anti-

mouse IgG antibodies were used as secondary antibodies to reveal UPF1-[295–914]-3×Flag and GAPDH signals. TrueBlot anti-mouse IgG secondary antibody coupled to HRP was used to reveal SMG6-[207–580]-HA signal.

**Analysis of the stability of endogenous NMD substrates by RT-qPCR.** These experiments were performed as previously described by Pr. Mühlemann's group [32,75]. Briefly, $5 \times 10^5$ HeLa cells expressing TCRβ WT reporter (kind gift from Pr. Mühlemann) were grown in 6-well plates in DMEM supplemented with 10% fetal bovine serum (FBS, Gibco) and 50 U/mL Penicillin and 50 mg/mL Streptomycin; DMEM+/+ at 37˚C and 5% $CO_2$ for 1 day (day 1). On day 2, cells were transfected by 500 ng of pSUPuro SMG6 and 250 ng of appropriate pcDNA3 plasmids (empty plasmid, p700; pMG1125, pMG1136, and pMG1137; S4 Table) using DreamFect transfection reagent (OZ Biosciences; #DF41000). On day 3, cells were transferred to DMEM+/+ supplemented with 1.5 µg/mL puromycin for 1 day in 6-cm plates. On day 4, puromycin containing medium was removed and fresh DMEM+/+ was added. On day 5, cells were collected and lysed to prepare total protein extracts for western blot analyses and to extract RNAs for further RT-qPCR.

To analyze the total proteins, the cells were resuspended in 50 mM Tris-HCl (pH 7.5), 150 mM NaCl, 1 mM EDTA, 1%(v/v) Triton X-100, 0.25%(w/v) Sodium Deoxycholate, and PIC 1×. After a 30-min incubation with slight shaking and centrifugation, 40 µg of total proteins were loaded on a 4% to 15% pre-casted protein gels (BioRad; #456–1084) prior to transfer on PVDF membrane for western blotting using anti-HA, anti-SMG6, or anti-GAPDH (loading control) antibodies (S6 Table).

Total RNAs were purified using the RNeasy Mini Kit (QIAGEN, #74134). First-strand cDNA was synthesized from purified RNA (1 µg) using the SuperScript III First-Strand Synthesis System (Thermo, #18080044) with a 50:50 mixture of oligo(dT)$_{20}$. Real-time qPCR was performed using iQTM SYBR Green Supermix (Bio-Rad, #1708882), primers at 300 nM (see S5 Table for primer sequences). *ACTB* (actin B) was used as endogenous control, and the comparative $C_T$ ($\Delta\Delta C_T$) method on a Bio-Rad CFX96 Real-Time PCR System and Bio-Rad Real-Time PCR PCR analysis Software (version 3.1).

## Supporting information

**S1 Fig. Comparison of Upf1 and Nmd4 structures.** (A) Superposition of the structure of Upf1-HD (color code indicated in the upper diagram and also used in panel B) bound to Nmd4 onto the structure of human UPF1-HD (light blue) bound to UPF2 (rmsd of 1.32 Å over 582 Cα atoms; PDB code: 2WJV; [53]). The largest difference between these structures is observed in the orientation of domain 1B. For the sake of clarity, yeast Nmd4, human UPF1-CH domain, and UPF2 region interacting with UPF1-CH have been omitted. (B) Superposition of the structure of Upf1-HD bound to Nmd4 onto the structure of human UPF1-HD (yellow) bound to ADP and phosphate (rmsd of 1.65 Å over 582 Cα atoms; PDB code: 2GK6; [52]). The largest difference between these structures is observed in the orientation of domain 1B. (C) Comparison of the crystal structures of *Kluyveromyces lactis* Nmd4 (blue; PDB code: 7QHY; [49]) and *S. cerevisiae* Nmd4 (as observed in our structure of the complex with Upf1-HD; the PIN domain and the arm are colored light and dark green, respectively; rmsd of 1.2 Å over 154 Cα atoms and 41% sequence identity). The amino acids matching with human SMG6 catalytic residues are shown as sticks. (D, E) Superposition of *S. cerevisiae* Nmd4 structure (as observed in our structure of the complex with Upf1-HD) onto human SMG5 (D; yellow; PDB code: 2HWY) or SMG6 (E; beige; PDB code: 2HWW) PIN domains (rmsd of 2 Å or 2.7 Å over 118 or 109 Cα atoms and 14% or 20% sequence identity, respectively; [34]). The amino acids matching with human SMG6 catalytic residues are shown as sticks. (F)

Comparison of the endonuclease active site of human SMG6 PIN domain (wheat) with the corresponding region from *S. cerevisiae* Nmd4 (light green). The amino acids corresponding to human SMG6 catalytic residues are shown as sticks.
(PDF)

**S2 Fig. Electron density 2mFo-DFc composite omit map illustrating the quality of the diffraction data.** The panels are centered around Nmd4 Nmd4 W216 (A) or F178 (B) residues. The maps (contoured at 1σ) have been calculated using the Phenix.composite_omit_map program implemented in the phenix.refine program suite [77,78]. For the sake of clarity, the electron density map is only shown on some specific residues of the interface.
(PDF)

**S3 Fig. Multiple sequence alignment of fungal Nmd4 orthologues.** Strictly conserved residues are in white on a black background. Partially conserved amino acids are shown in bold. Secondary-structure elements, as observed in the crystal structure of the *S. cerevisiae* Nmd4 protein bound to Upf1-HD, are shown above the alignment. Positions involved in the interface with Upf1-HD are indicated by black filled spheres below the alignment. Residues R210 and W216 mutated in this study are boxed in red. Domain boundaries are indicated above the alignment using the color code defined in Fig 1A. This figure was generated using the ENDscript server [76].
(PDF)

**S4 Fig. Multiple sequence alignment of Upf1 proteins.** Strictly conserved residues are in white on a black background. Partially conserved amino acids are shown in bold. Secondary-structure elements, as observed in the crystal structure of the *S. cerevisiae* Upf1-HD protein bound to Nmd4, are shown above the alignment. Positions involved in the interface with Upf1-HD are indicated by black filled spheres below the alignment. The yeast Upf1 residues G243 and G377 mutated in this study are boxed in red. Domain boundaries are indicated above the alignment using the color code defined in Fig 1A. This figure was generated using the ENDscript server [76].
(PDF)

**S5 Fig. The « arm » region from Nmd4 is sufficient for the interaction with Upf1-HD.** (A) The Nmd4 « arm » domain is sufficient for the interaction with Upf1-HD in vitro. Pull-down experiments of Upf1-HD by different His-tagged Nmd4 domains. Input and elution (His) samples were analyzed on 12% SDS/PAGE and Coomassie blue staining. L: Molecular weight ladder. (B) The Nmd4 « arm » region is required for its interaction with Upf1 in vivo. Co-purification of Nmd4-FL-HA or Nmd4-PIN-HA with Upf1-FL-TAP was evaluated using an anti-HA immunoblot on total extracts and purified fractions. Strains expressing Nmd4-FL-HA and Nmd4-PIN-HA and native untagged Upf1 were used as negative controls.
(PDF)

**S6 Fig. Effect of Nmd4 on Upf1-HD RNA binding activity.** Interaction of Upf1-HD, $His_6$-ZZ-Nmd4-FL, $His_6$-ZZ-Nmd4-FL/Upf1-HD complex, and $His_6$-ZZ with RNA poly(U)$_{30}$ studied by ITC. The data underlying this figure can be found in S6 Data.
(PDF)

**S7 Fig. Gel filtration chromatograms of *S. cerevisiae* Upf1-HD (A) or Nmd4 (B) proteins used in our in vitro assays.** (A) Size exclusion chromatography (S200 16/60) profiles obtained for pure Upf1-HD WT and mutant proteins, showing very similar retention times for these different proteins. The Coomassie stained SDS-PAGE analysis of the proteins present in the main peak is shown for every protein (WT and mutants) to show the purity of the final

proteins. MW: Molecular weight ladder. (B) Size exclusion chromatography (S75 16/60) profiles obtained for pure His$_6$-ZZ-Nmd4 WT and mutant proteins, showing very similar retention times for these different proteins. The Coomassie stained SDS-PAGE analysis of the proteins present in the main peak is shown for every protein (WT and mutants) to show the purity of the final proteins. MW: Molecular weight ladder. The data underlying this figure can be found in S7 Data.

(PDF)

**S8 Fig. Control figure (immunoblot TAP-Upf1 variants).** The amounts of expressed UPF1 variants from plasmids were estimated by immunoblot against the N-terminal TAP tag. Each sample was diluted to one half and one fourth, to estimate the quantitative aspect of the immunoblot.

(PDF)

**S9 Fig. AlphaFold3 model of the complex between human SMG6 and UPF1-HD.** (A) Superposition of the 5 AF3 models of the UPF1-HD/SMG6-[398–494] complex. For the sake of clarity, only 1 UPF1-HD model is shown in blue. The SMG6 fragments are colored differently depending on the models. The coordinates of the 5 AF3 models are provided as S1–S5 Files. (B) Overview of the best AF3 model of the human SMG6 [398–494] region bound to UPF1-HD domain, colored by pLDDT values. For the sake of clarity, only SMG6 residues 438 to 480 are shown. (C) Detailed view of the interface between human UPF1-HD (same color code as Fig 1B) and SMG6-[398–494] (colored by pLDDT values) in the best AF3 model. Residues from human SMG6 shown as sicks are in orange and underlined. (D) Predicted aligned error (PAE) plot of the prediction of the UPF1-HD/SMG6-[398–494] complex. This panel was generated using the PAE Viewer website (https://subtiwiki.uni-goettingen.de/v4/paeViewer Demo), the S1 File as structure file and S6 File as scores file. (E) Superposition of AF3 model of the human SMG6 [398–494] region (orange) bound to UPF1-HD domain (omitted for the sake of clarity) onto the yeast Nmd4/Upf1-HD crystal structure (same color code as Fig 1B). The side chains of R210 and W216 from Nmd4 (underlined labels) and R448 and W456 (labels underlined and in italics) of SMG6 are shown as sticks. The Cα atoms of yeast Upf1 Gly243 and Gly377 are shown as spheres.

(PDF)

**S10 Fig. Control figure showing the silencing of endogenous SMG6 and the rescue by the expression of shRNA-resistant variants of SMG6.** Total extracts were loaded on SDS-PAGE and proteins were detected by western blot using the indicated antibodies.

(PDF)

**S1 Table. Statistics of data collection and structure refinement.**

(XLSX)

**S2 Table. Details of the electrostatic interactions involved in the Nmd4/Upf1-HD interface.**

(XLSX)

**S3 Table. Oligonucleotides used in this study.**

(XLSX)

**S4 Table. Plasmids used in this study.**

(XLSX)

**S5 Table. Yeast strains used.**
(XLSX)

**S6 Table. Antibodies used in this study.**
(XLSX)

**S7 Table. Primers used for RT-qPCR experiments.**
(XLSX)

**S1 Data. Raw data and processed data for the ITC experiments shown in Fig 2A.**
(XLSX)

**S2 Data. Raw data and processed data for the ATPase assays shown in Fig 2B.**
(XLSX)

**S3 Data. Raw data and processed data for the RT-qPCR experiments shown in Fig 3E.**
(XLSX)

**S4 Data. Raw data and processed data for the ATPase assays shown in Fig 3G.**
(XLSX)

**S5 Data. Raw data and processed data for the RT-qPCR experiments shown in Fig 4C.**
(XLSX)

**S6 Data. Raw data and processed data for the ITC experiments shown in S6 Fig.**
(XLSX)

**S7 Data. Raw data for the SEC profiles shown in S7 Fig.**
(XLSX)

**S1 Raw Images. Raw images of original gels and blots.**
(PDF)

**S1 File. PDB coordinates of the first model for the complex between human UPF1-HD and SMG6-[398–494] region generated by AlphaFold (v3).**
(PDB)

**S2 File. PDB coordinates of the second model for the complex between human UPF1-HD and SMG6-[398–494] region generated by AlphaFold (v3).**
(PDB)

**S3 File. PDB coordinates of the third model for the complex between human UPF1-HD and SMG6-[398–494] region generated by AlphaFold (v3).**
(PDB)

**S4 File. PDB coordinates of the fourth model for the complex between human UPF1-HD and SMG6-[398–494] region generated by AlphaFold (v3).**
(PDB)

**S5 File. PDB coordinates of the fifth model for the complex between human UPF1-HD and SMG6-[398–494] region generated by AlphaFold (v3).**
(PDB)

**S6 File. Scores file of the first model for the complex between human UPF1-HD and SMG6-[398–494] region provided by AlphaFold (v3) and used to generate S9E Fig.**
(JSON)

## Acknowledgments

We thank Clara Moch, Bérénice Pénicaut, Lucia Oreus, and Magali Aumont-Nicaise for technical assistance. We are indebted to Pr Oliver Mühlemann (University of Bern, Switzerland) for kindly sharing with us reagents. We are grateful to the staff at the SOLEIL Synchrotron, France (proposal Nos. 20181001 and 20201046), in particular, Martin Savko, William Sheppard, and Serena Sirigu (PROXIMA-2A beamline), for smoothly running the facility. CS and LD thank Alain Jacquier and Guilhem Janbon for support and stimulating discussions and Lucia Oreus for media and buffers preparation.

## Author Contributions

**Conceptualization:** Irène Barbarin-Bocahu, Hervé Le Hir, Cosmin Saveanu, Marc Graille.

**Funding acquisition:** Hervé Le Hir, Cosmin Saveanu, Marc Graille.

**Investigation:** Irène Barbarin-Bocahu, Nathalie Ulryck, Amandine Rigobert, Nadia Ruiz Gutierrez, Laurence Decourty, Mouna Raji, Bhumika Garkhal, Cosmin Saveanu, Marc Graille.

**Writing – original draft:** Marc Graille.

**Writing – review & editing:** Irène Barbarin-Bocahu, Nathalie Ulryck, Nadia Ruiz Gutierrez, Bhumika Garkhal, Hervé Le Hir, Cosmin Saveanu, Marc Graille.

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
