## [Editor Report · Decision Letter 0]

16 Apr 2024

Dear Dr Graille, 

Thank you for submitting your revised manuscript from Review Commons entitled "Identification of an evolutionary conserved binding motif responsible for the recruitment of NMD factors to the UPF1 helicase" for consideration as a Research Article by PLOS Biology. Please accept my sincere apologies for the delay in getting back to you as we consulted with an academic editor about your submission, especially given the competing paper noted in your cover letter. We will do our utmost to ensure a quick review process from this point onwards. 

Your manuscript has now been evaluated by the PLOS Biology editorial staff, as well as by an academic editor with relevant expertise, and I am writing to let you know that we would like to send your submission back out to the original reviewers at Review Commons. 

However, before we can send your revised manuscript to back out to the reviewers, we need you to complete your submission by providing the metadata that is required for full assessment. To this end, please login to Editorial Manager where you will find the paper in the 'Submissions Needing Revisions' folder on your homepage. Please click 'Revise Submission' from the Action Links and complete all additional questions in the submission questionnaire.

Once your full submission is complete, your paper will undergo a series of checks in preparation for peer review. After your manuscript has passed the checks it will be sent out for review. To provide the metadata for your submission, please Login to Editorial Manager (https://www.editorialmanager.com/pbiology) within two working days, i.e. by Apr 18 2024 11:59PM.

Kind regards,

Richard

Richard Hodge, PhD

rhodge@plos.org

PLOS

---

## [Decision Letter · Decision Letter 1]

16 May 2024

Dear Marc,

Thank you for your continued patience while we considered your revised manuscript from Review Commons entitled "Identification of an evolutionary conserved binding motif responsible for the recruitment of NMD factors to the UPF1 helicase" for consideration as a Research Article at PLOS Biology. Please accept my sincere apologies for the delays that you have experienced during this round of the peer review process, especially given the competing manuscript. Your revised study has now been evaluated by the PLOS Biology editors, the Academic Editor and the original reviewers at Review Commons. 

As you will see in the reviews (pasted below), the reviewers are largely positive about the revised manuscript and are supportive of publication. Reviewer #1 asks that additional reporting details are included for the AlphaFold modelling and Reviewer #2 requests that additional discussions are included regarding the inconsistency of nuclease activity. In addition, I have included some additional comments from the Academic Editor handling your submission (labelled ‘Comments from the Academic Editor’) who has noted that a few important controls are missing in the revision that were not raised in the previous round at Review Commons. We ask that you please include these controls in the revised version, including a control for total protein mass in the ATPase assays in Figure 2 and 3, as well as controls to confirm that the effects seen in Figure 3E are dependent on Upf1-Nmd4 binding. 

Therefore, we are pleased to offer you the opportunity to address the remaining points from the reviewers and Academic Editor in a revision that we anticipate should not take you very long. We will then assess your revised manuscript and your response to the reviewers' comments with our Academic Editor to avoid further rounds of peer-review.

In addition, after further discussions with the editorial team, we would like to consider your manuscript as a Short Report at the journal (https://journals.plos.org/plosbiology/s/what-we-publish#loc-short-reports) given the shorter format of the manuscript. Upon resubmission, I would be grateful if you could please tick 'Short Report' as the article type in the dropdown menu. 

We expect to receive your revised manuscript within 1 month, but please let us know if you need more time. Please email us (plosbiology@plos.org) if you have any questions or concerns, or would like to request an extension. 

**IMPORTANT - SUBMITTING YOUR REVISION**

*Resubmission Checklist*

*Published Peer Review*

*PLOS Data Policy*

*Blot and Gel Data Policy*

Sincerely,

Richard

Richard Hodge, PhD

rhodge@plos.org

REVIEWS:

Reviewer #1: The authors have adequately addressed the points I raised in my previous review. I believe that the manuscript is suitable for publication.

My remaining concern is about the included Alphafold model in Fig EV6. 

This figure also needs to show the Alphafold confidence scores, otherwise readers are unable to appreciate whether the model can be trusted. 

- One panel should show the model colored according to the pLDDT score.

- If the SMG6 region binding to UPF1 is not clearly visible in the pLDDT-coloured figure, it should also be shown on its own with highlighting UPF1-binding residues as sticks (pLDDT- colored).

- A PAE diagram for the complex prediction should also be shown

- A panel showing the predicted interaction details could be shown.

Reviewer #2: The authors have addressed most of my comments. 

With regard to the limitation highlighted earlier, please provide a brief discussion of the consistency of phosphorylation-independent binding versus the inconsistency of nuclease activity.

Please revise the sentence "but also that a physical interaction between Upf1-HD and the PIN domain exists in vitro" to "might exist".

COMMENTS FROM THE ACADEMIC EDITOR

1. This work heavily relies on the use of in vitro assays to determine the role of Nmd4-Upf1 protein-protein interactions on NMD function (ATPase assays in Figure 2/3, binding assays in Figure 3/4). Though it is encouraging, for example, that Nmd4 FL, PIN, and Arm purified proteins differentially promote Upf1 ATPase activity in Figure 1B, it is concerning that all additions increase assay output. This is especially the case given that the Nmd4 PIN domain was not shown to bind Upf1 yet still increases its ATPase activity. A control for total protein mass in the ATPase assay should be included to address this possible confounding effect, perhaps with a relevant protein without known Upf1-HD binding. 

2. If the effects seen in Figure 3E, with respect to processing of DAL7 and pre-L28, are truly dependent on Nmd4 binding to the Upf1-HD-Ct protein, deletion of Nmd4 in the background of Upf1-HD-Ct G243R and G377R should demonstrate no change from expression of these constructs in an NMD4 strain. These two controls (Upf1-HD-Ct G243R and G377R in a upf1nmd4 background) should be added to confirm that the effects are dependent on Upf1-Nmd4 binding. Furthermore, results in 3F suggest that Nmd4 R210E, W216A, and M2 should act similarly to nmd4 in the experiments from 3E, which would be important controls to validate in vivo the in vitro findings from other panels in Figure 3.

---

## [Editor Report · Decision Letter 2]

19 Aug 2024

Dear Marc,

Thank you for your patience while we considered your revised manuscript "Identification of an evolutionary conserved binding motif responsible for the recruitment of NMD factors to the UPF1 helicase" for publication as a Research Article at PLOS Biology. This revised version of your manuscript has been evaluated by the PLOS Biology editors and the Academic Editor.

Based on our Academic Editor's assessment of your revision, I am pleased to say that we are likely to accept this manuscript for publication, provided you address the following data and other policy-related requests that I have provided below (A-H):

(A) After discussions with the editorial team, we think your manuscript would be a better fit as a Short Report. Upon resubmission, please tick 'Short Report' as the article type in the drop down menu in the online submission form. 

(B) We would like to suggest the following modification to the title:

“Structure of the Nmd4-Upf1 complex supports conservation of the nonsense-mediated mRNA decay pathway between yeast and humans”

(C) Please move the figures labelled ‘EV’ into the supplementary figure file (S1 Fig).

(D) You may be aware of the PLOS Data Policy, which requires that all data be made available without restriction: http://journals.plos.org/plosbiology/s/data-availability. For more information, please also see this editorial: http://dx.doi.org/10.1371/journal.pbio.1001797

-Supplementary files (e.g., excel). Please ensure that all data files are uploaded as 'Supporting Information' and are invariably referred to (in the manuscript, figure legends, and the Description field when uploading your files) using the following format verbatim: S1 Data, S2 Data, etc. Multiple panels of a single or even several figures can be included as multiple sheets in one excel file that is saved using exactly the following convention: S1_Data.xlsx (using an underscore).

-Deposition in a publicly available repository. Please also provide the accession code or a reviewer link so that we may view your data before publication. 

Figure 2A-B, 3E, 3G, 4C, EV5, EV6

(E) Please also ensure that each of the relevant figure legends in your manuscript include information on *WHERE THE UNDERLYING DATA CAN BE FOUND*, and ensure your supplemental data file/s has a legend.

(F) Thank you for already providing the original and uncropped gel/blot images in the ‘Uncropped gels’ file. However, we note that the images for the following figures are missing in this document:

Figures S2A-B, S3, S4

We will require these files before the manuscript can be accepted so please prepare and upload them now. Please carefully read our guidelines for how to prepare and upload this data: https://journals.plos.org/plosbiology/s/figures#loc-blot-and-gel-reporting-requirements

(G) Please ensure that your Data Statement in the submission system accurately describes where your data can be found and is in final format, as it will be published as written there. 

(H) Per journal policy, if you have generated any custom code during the course of this investigation, please make it available without restrictions. Please ensure that the code is sufficiently well documented and reusable, and that your Data Statement in the Editorial Manager submission system accurately describes where your code can be found. 

We expect to receive your revised manuscript within two weeks. 

*Published Peer Review History*

*Press*

Kind regards,

Richard

Richard Hodge, PhD

rhodge@plos.org

PLOS

---

## [Editor Report · Decision Letter 3]

29 Aug 2024

Dear Marc,

On behalf of my colleagues and the Academic Editor, Jeff Coller, I am pleased to say that we can accept your manuscript for publication, provided you address any remaining formatting and reporting issues. These will be detailed in an email you should receive within 2-3 business days from our colleagues in the journal operations team; no action is required from you until then. Please note that we will not be able to formally accept your manuscript and schedule it for publication until you have completed any requested changes.

PRESS

Best wishes, 

Richard

Richard Hodge, PhD

rhodge@plos.org

PLOS
